# Deep learning and superoscillatory speckles empowered multimode fiber probe for in situ nano-displacement detection and micro-imaging

Lele Wang [1,2,5], Yiwei Zhang[3,5], Yibing Zhou [3], Yuan Meng [1], Zhengyang Lu[4], Pei Li[1,2], Hailong Zhang[1,2], Dan Li[1,2], Ping Yan [1,2], Qirong Xiao [1,2] ✉ & Qiang Liu[1,2] ✉

High-precision metrology has laid the foundation for semiconductor fabrication and life sciences. However, existing displacement measurement approaches are incapable of performing flexible probing within complex equipment interiors. Here, we present a in situ, non-contact nano-displacement measurement approach. Leveraging a multimode fiber probe empowered by deep learning, fine feature information can be efficiently extracted from superoscillatory speckles, achieving single-ended detection with 10 nm resolution and 99.95% accuracy. A physical model is established to correlate the displacement with higher-order modes proportion in the fiber. Sub-millimeter-sized probe enables detecting targets with different structures in confined spaces. Robust recognition is achieved through joint learning, under varying fiber bending conditions and different metal materials. With extreme compression ratios of less than 0.1%, the system delivers high accuracy, low training costs, and high-speed processing. The imaging capability of the probe is also experimentally validated, proving potential as a powerful tool in applications such as lithography, weak force sensing, and super-resolution micro-endoscopy.

Precise metrology plays a crucial role in both the exploration of natural sciences[1,2] and the advancement of engineering sciences[3]. The birth and advancement of laser technology enable high-precision displacement measurement instruments for various technologies such as gravitational wave observation in general relativity[4], and lithography in semiconductor manufacture[5]. Technologies like femtosecond optical frequency combs[6], superoscillatory optical ruler[7], metasurface[8,9], photonic gears[10], and single-cavity loss[11] continuously break the limits of displacement measurement, bringing the resolution of displacement measurements into the sub-wavelength or even nanometer scale.

These technologies typically achieve enhanced measurement precision by constructing ratios of signals with different periodic physical quantities, already showing extraordinary potential in fields such as super-resolution microscopy[12], weak force sensing[13], particle tracking[14], and precision mechatronic equipment[15], particularly in the realms of nano-monitoring and nano-manufacturing.

Currently, nano-displacement measurement mainly relies on high-performance laser interferometers, which can suppress noise below the classical limit. However, these Michelson/Mach-Zehnder setups[16], including white-light interferometers, are extremely

¹Department of Precision Instrument, Tsinghua University, Beijing, China. ²State Key Laboratory of Precision Space-time Information Sensing Technology, Beijing, China. ³Weiyang College, Tsinghua University, Beijing, China. ⁴Weixian College, Tsinghua University, Beijing, China. ⁵These authors contributed equally: Lele Wang, Yiwei Zhang. ✉e-mail: xiaoqirong@mail.tsinghua.edu.cn; qiangliu@tsinghua.edu.cn

susceptible to interference from airflow, platform vibrations, and electromagnetic noise[17]. To maintain measurement accuracy, high requirements are placed on the measurement environment. Additionally, the complex optical path design and the need for stable light sources[18] make it difficult to reduce both technological and economic costs. On the other hand, due to spatial constraints, compact longitudinal displacement monitoring devices[19,20] are particularly important in semiconductor industry and super-resolution endoscopy. Traditional optical interferometers are bulky and require optical path deflection components in the monitoring environment, making them difficult to deploy for flexible detection. Although eddy current sensors[21] and capacitive probes[22] can achieve remote detection, their probe sizes are still limited to tens of millimeters, which restricts their use to conductive surfaces. Internal integrated displacement sensors[20] often require contact with the detected target, posing a risk of damaging microstructures[23]. Novel technologies such as orbital meta-atoms[24] and single-cavity loss[11] demonstrate high resolution and high signal-to-noise ratios, but they require access to both sides of the detection target, which is difficult to achieve in many scenarios. Non-contact single-ended access for in situ nano-displacement measurement remains a significant challenge.

Multimode fiber (MMF), as a commonly used optical waveguide material, can support thousands of transverse optical field modes. Coupled with its flexible and small physical properties, it has recently been actively applied in fields such as ultra-thin endoscopy[25–27], deep brain neural observation[28,29], nonlinear optical computing[30], and micro-spectrometers[31]. Due to mode dispersion and intermodal coupling effects, the phase and intensity ratios of these transverse mode distributions dramatically change during transmission. It can be seen as disordered media with strong scattering, resulting in high-density speckle formation through coherent superposition at the fiber's output face. Speckle fields have demonstrated high sensitivity to transverse displacement[32,33], spectral shifts[34], temperature[35], and other physical quantities. However, the potential of multimode fiber systems in phase detection or displacement sensing has not been fully explored. On the other hand, the rapid development of deep learning has introduced new opportunities for traditional optical metrology[36]. Data-driven approaches offer novel solutions to ill-posed inverse problems. However, they are mainly applied to image-processing-related deep stereo matching[37] and phase reconstruction[38]. The full potential of deep learning in handling small parameters and processing multi-tasks has yet to be fully realized.

Here, we propose and experimentally validate a non-contact in situ nano-displacement measurement scheme. By introducing an ultrathin (800 μm) multimode fiber probe and the Visual Geometry Group (VGG) convolutional neural network architecture, the single-end access system achieves high-resolution (10 nm) and high-accuracy (99.95%) extraction of wafer displacement information through the device's original narrow slit. From physical perspective, we reveal the relationship between higher-order transmission mode components of the fiber and the detected displacement. We discovered super-oscillation in the multimode fiber speckle field, demonstrating the capability of this strong scattering medium in the field of super-diffraction-limit optical metrology. The all-fiber-integrated system achieves lens-free portable access with fiber-end-ball, side-coupler, and other specialized fiber-optic devices. Through joint learning, we achieve displacement measurement for different bending states of the probe and various metallic materials recognition, exhibiting high robustness and multi-task processing capability. By compressive sampling of the speckle field with 0.28‰ of information, we achieve positioning under small dataset size and small parameters training, reducing computational costs and improving computation speed. Moreover, we experimentally validate the imaging capabilities of the fiber probe. By introducing an inverse transmission matrix (ITM) optimization network, we enable simultaneous longitudinal

displacement measurement and image reconstruction. By extracting ITMs at different positions to construct a matrix library, we optimize imaging quality. This integrated fiber detection system, combining high-precision metrology and imaging, is expected to have advantages in photolithography in confined spaces, weak force sensing, and implantable flexible microendoscopes.

## Results

### In situ nano-displacement fiber probing scheme

Multimode fibers have demonstrated excellent performance in the metrology of various physical quantities such as spectrum[31,34,39], pressure[40], and temperature[35,41]. These waveguides, typically with hundreds of micrometers diameters, can support thousands of transverse optical field distributions and have the ability to detect and transmit optical signals at the same time. In step-index fibers, mode dispersion effects caused by different propagation speeds of modes and inter-mode coupling effects resulting from uneven refractive index distributions distort the input optical field. This distortion leads to the output of a high-density speckle field formed by the coherent superposition of multiple modes. Changes in the physical quantities cause small variations in the input optical signal's phase, frequency, polarization, and intensity, or even changes in the fiber's morphology. These variations lead to changes in the evolution of transverse modes, which accumulate as the light propagates along the fiber, ultimately resulting in changes in the speckle patterns. By constructing the relationship between the acquired speckle field and the physical quantities, precise detection or imaging can be achieved.

Figure 1 illustrates the architecture of the proposed multimode fiber-based detection system for wafer longitudinal displacement measurement within a semiconductor processing system. The detection light is introduced into the multimode fiber probe via a side-coupler device. With a probe size of 800 μm, the fiber enables in situ detection inside the system through narrow slits or tubes, without the need to fully open the system. The light output from the probe is collimated via a fiber-end-ball fabricated at the fiber end face and directed onto the wafer. The light is then reflected by the wafer, received again by the fiber-end-ball, and transmitted back through the MMF. The output speckle field is captured by a two-dimensional array sensor (Detailed structure of the system and fiber configuration can be found in Supplementary Fig. S21). Figure 1b shows the distributions and characteristics of speckles at four different displacements ranging from 0 to 300 nm. Below the speckle patterns, the grayscale histogram and intensity variation along the transverse diameter tangent are plotted. It is evident that even nanometer-scale displacement changes lead to noticeable shifts in speckle spot positions and variations in the statistical distribution. Supplementary Movie 1 demonstrates speckle field varies in real time with wafer displacement. This sensitivity of the speckle pattern to displacement enables high-precision displacement measurement. Using the VGG convolutional neural network under deep learning, we achieved displacement measurement with a resolution of 10 nm and an accuracy of 99.95% for different structured patterns.

### Physical relationship between displacement and high-order modes proportion

To explore the effect of the longitudinal displacement of the detected target on the output speckle field of a multimode fiber, we analyze the higher-order transverse mode ratios at different displacements. A MMF physical model is developed to simulate the propagation and coupling process of the returned signal light from the detected target by calculating the input optical field evolution of step-index MMF and mode decomposition[42]. The mode decomposition coefficients of the same input light field at different detection distances are obtained by Fresnel integration, spherical lens transformation, and Cartesian

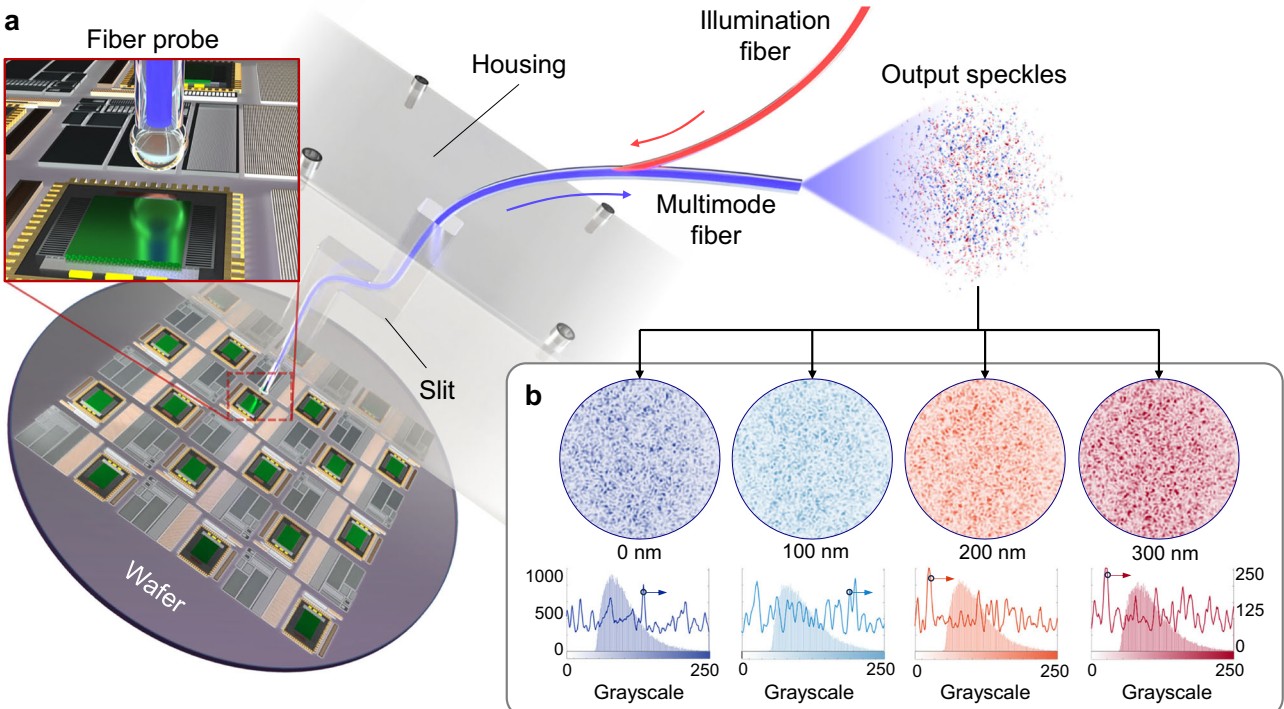

**Fig. 1 | In situ nano-displacement measurements using a multimode fiber probe. a** Displacement probing of wafers inside the device by inserting the multimode fiber probe through the slit, eliminating the need for an open enclosure. An integrated fiber-end-ball at the detection end helps to homogenize and collimate the beam. **b** Output speckle field distribution as well as its statistical characteristics of the multimode fiber for wafers at 0 nm, 100 nm, 200 nm, and 300 nm displacements. Below the speckles are the respective grayscale histograms (corresponding to the left axis) and the intensity fluctuation curves of the transverse diameters (corresponding to the right axis), reflecting a clear differentiation.

coordinate basis to linear polarization (LP) mode coordinate basis conversion.

Figure 2a shows the physical simulation model. Figure 2b presents the input optical field chosen for the simulation. Since the output speckle field is also influenced by the two-dimensional spatial structure of the detected target, we selected a wafer micrograph, which contains both high-frequency and low-frequency structures, to excite higher-order modes and obtain universal results. Figure 2c–d shows the fiber input facet and field at the core area after free space transmission and fiber-end-ball transformation, respectively. We performed mode decomposition on the input optical field and selected the first 1000 transverse mode components for reconstruction, as shown in Fig. 2e. It can be observed that Fig. 2e contains most of the structures from Fig. 2d. We further analyze the energy percentage of the mode components in the input optical field and their relationship with the displacement. The first 1000 modes are divided into 10 groups in sequential order, with each group containing 100 modes. The intensity percentage of each group is the sum of the intensity percentages of the 100 modes in the group. As the detected displacement increases from 0 mm to 30 mm, it can be found from Fig. 2f that the share of the fundamental mode component gradually increases, and more energy is converted from higher-order modes to lower-order modes. This means that within the large-scale range of 0–30 mm the farther the detected target is from the fiber probe, the fewer high-order mode components in the speckle field, corresponding to a decrease in the intensity of the speckle periphery. Figure 2g shows the percentage of the first 20 modes and the distribution of the main modes at the detected displacement of 0 nm. It can be observed that LP patterns exhibiting rotational symmetry (LP$_{0n}$) generally account for a higher proportion of intensity.

In order to visually characterize the influence of the displacement on the transmission modes in MMFs, the normalized average mode order is defined:

$$O_n = \frac{\sum_i P_l(i) \times i}{\sum_i P_l(i)} \tag{1}$$

which is the product of the order $i$ of the different modes generated after coupling the optical field into the MMF and their corresponding energy occupancy coefficients $P_l(i)$, representing the average mode order of the coupled mode combination. This coefficient reflects to some extent the relationship between the displacement and the output speckle distribution. The result in Fig. 2h shows that as the displacement distance increases, the normalized average mode order shows a monotonically increasing trend, while the proportions of both the first 100 modes and the first 1000 LP modes demonstrate monotonically decreasing trends. The law observed here is opposite to that in Fig. 2f. It may be attributed to the detection target not reaching the equivalent focal plane of the fiber-end-ball. Partial defocusing effects will cause a reduction of high-frequency components[43], resulting in a higher proportion of fundamental modes. As the displacement distance increases, the target approaches the equivalent focal plane of the fiber-end-ball. The energy of the coupled optical field shifts toward higher-order modes. To further investigate the patterns under large displacement ranges, we have supplemented the simulation results of wafer microscopic images within a large displacement measurement range of 0–10 mm in the Supplementary Information. In this case, the results are consistent with the laws of Fig. 2f. More details of the physical model and results are presented in Supplementary Note 1, Supplementary Table 1, and Supplementary Fig. S1, S2 and S3.

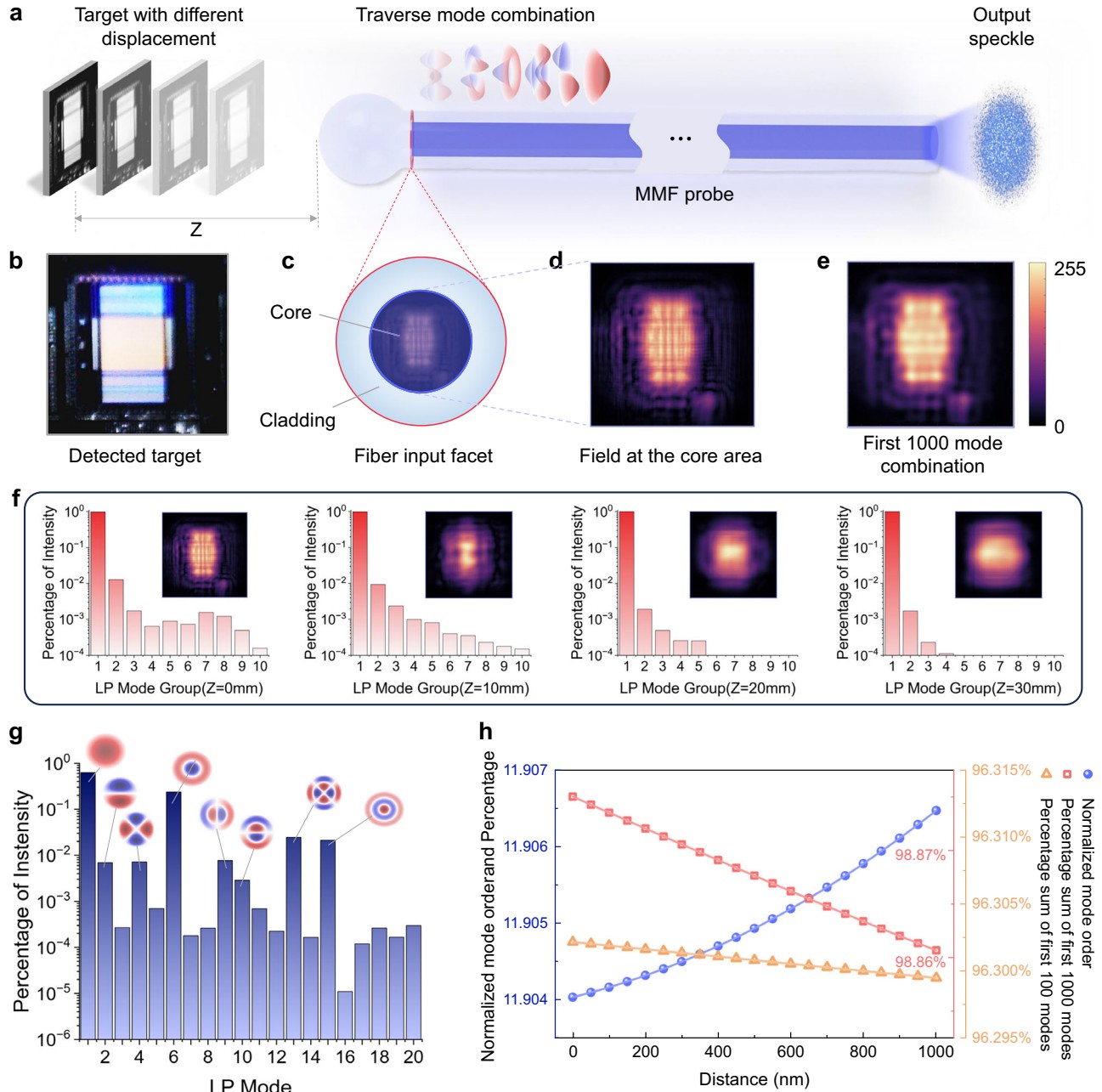

**Fig. 2 | Physical model of MMF and simulation results at different displacements. a** Different combinations of transmission modes of MMFs are coupled at different detected displacement z, forming different speckles. **b–e** The detected target, fiber input facet, field at the core area, and the recombination result of the first 1000 modes. **f** The first 1000 modes are divided into 10 groups in sequential order, with each group containing 100 modes. The intensity percentage of each group is the sum of the intensity percentages of the 100 modes in the group. The intensity percentage of these ten groups and the combination field at 0 mm, 10 mm, 20 mm, and 30 mm displacements. **g** Percentage of the first 20 modes and the distribution of the main modes at the detected displacement of 0 nm. **h** Normalized average mode order (blue left axis), percentage of the first 100 modes (orange right axis) and the first 1000 modes (red right axis) at different detected displacement (0–990 nm, 100 sampling points).

## Displacement detection joint learning method

The above analysis reveals the underlying mechanism by which displacement influences the input light field and the distribution of higher-order modes within the fiber. This leads to changes in the output speckle pattern. To explore the resolution capability of multi-mode fiber speckle fields in displacement measurement, we used the Transport of Intensity Equation method[44] (see Supplementary Note 2 and Supplementary Fig. S4) to obtain the phase and phase gradient distribution of the speckle field. Figure 3a shows the distribution and zoomed-in view of the speckle phase gradient modulus $|k| = |\Delta\varphi|$. We clearly observe nano-scale phase discontinuity patterns below the $\lambda/2$ level. This rapid phase variation, or phase singularity, is characteristic of speckle superoscillation[45], which has been demonstrated to possess sub-nanometer displacement resolution capability[33]. We believe that the multiple interference effects between the numerous different mode wavefronts within the multimode fiber lead to such sub-wavelength free-space singular light fields, enabling measurements beyond the traditional diffraction limit.

Traditional speckle metrology methods rely on comparing the similarity of the collected speckles with the speckles under the target scale. We used the multimode fiber probe to detect the displacement of a Digital Micromirror Device (DMD) to construct a validation

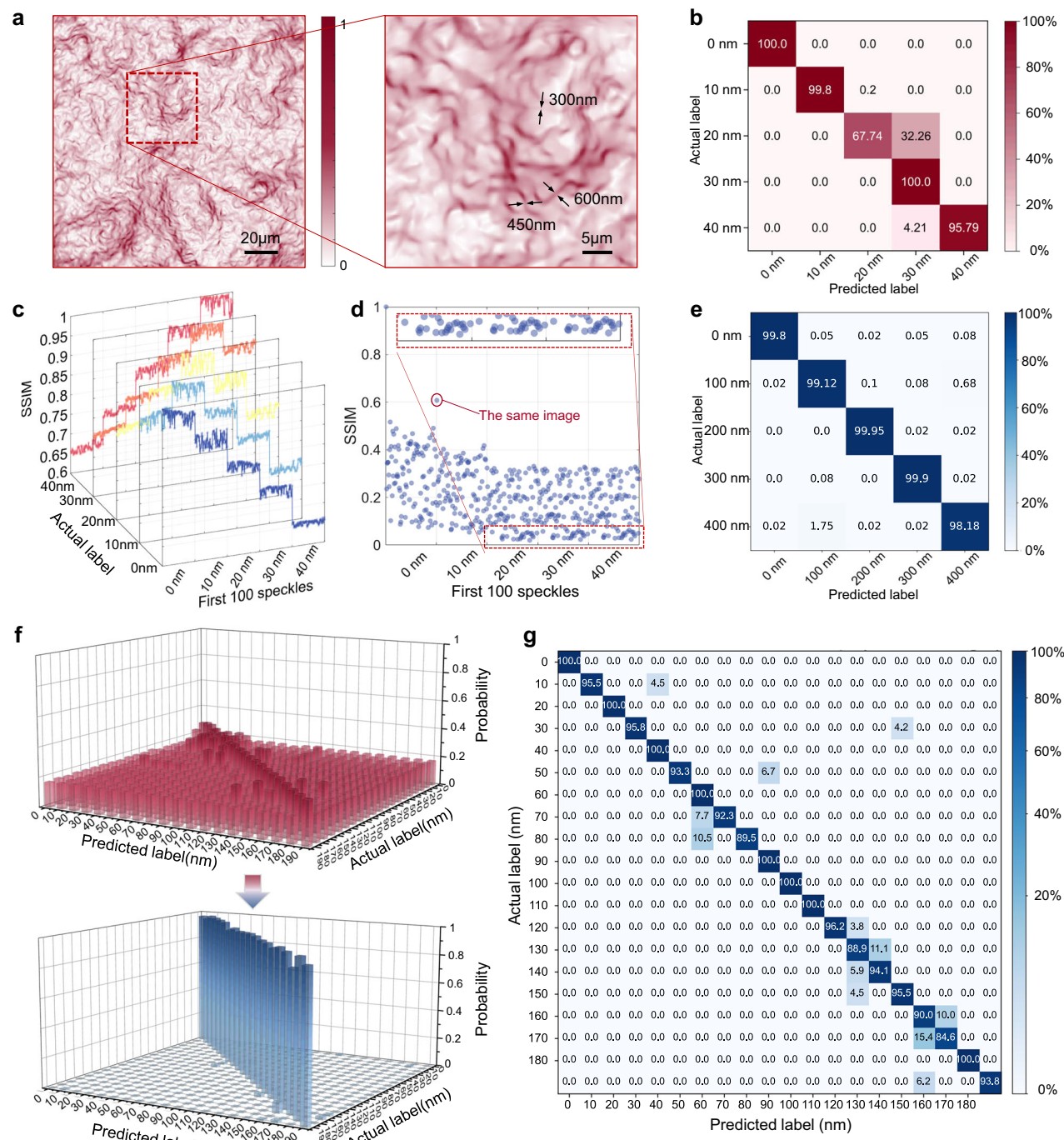

**Fig. 3 | Comparison results between traditional speckle similarity measurement and deep learning methods. a** Speckle phase gradient distribution obtained by utilizing the Transport of Intensity Equation and its local magnification. It contains structures with fast phase changes with a hundred-nanometer size. **b** Displacement identification confusion matrix obtained by detecting target with the same structure at 10 nm resolution using the speckle similarity comparison method. The horizontal axis is the displacement recognized by the system and the vertical axis is the actual displacement. Numbers in the grid are percentage probabilities. Values on the diagonal represent correct recognition. **c** The similarity between the speckles acquired at different displacements and the labeled speckles

(first speckles at 0 nm, 10 nm, 20 nm, 30 nm, and 40 nm) with the target's structure unchanged. **d** The similarity between the speckles collected at different displacements and the labeled speckle (first one at 0 nm) with the detected target's structure cyclically changing. **e** Confusion matrix for bidirectional displacement recognition at 100 nm resolution for targets with different structures using deep learning methods. **f** Displacement confusion matrix for a varying structural target with 20 sets of samples at 10 nm resolution using speckle similarity comparison (upper) and joint learning methods (lower). **g** Confusion matrix for measurements at a resolution of 10 nm and 20 groups per 100 datasets size.

experiment. Figure 3b shows the measurement results of displacement at a 10 nm resolution, using the speckle similarity comparison method when the DMD displays a uniform white pattern. For the range of 0–40 nm, data was collected at five displacement positions, with 500

speckle images captured at each position. The first speckle patterns at each position were taken as labels, and the remaining speckles were used for testing. The structural similarity (SSIM) was calculated for each of the five label speckles, and the label with the maximum

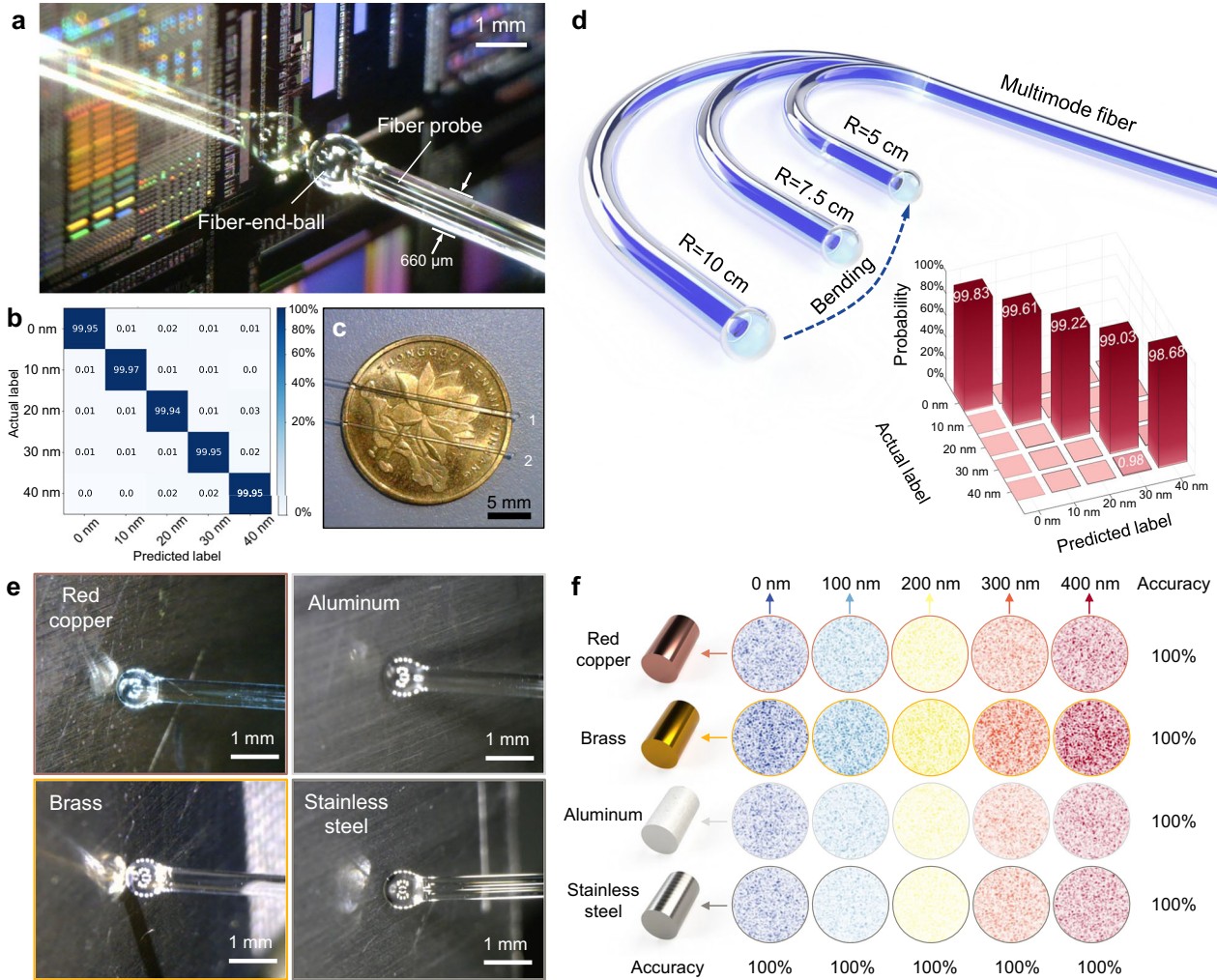

**Fig. 4 | Experimental results of detecting wafer and different metal plates.**
**a** Micrographs of displacement detection of a wafer using the multimode fiber probe. **b** Displacement confusion matrix for wafer with 10 nm resolution in 0−40 nm range. **c** Size comparison of two different diameters of multimode fiber probes with a 50-cent Chinese coin. **d** Fiber probe detection under different bending states. Confusion matrix for joint learning at three bending radii: 5 cm, 7.5 cm, 10 cm. **e** Micrographs of four different metal materials (red copper, brass, aluminum, and stainless steel) detected by the fiber probe. **f** Identification results of different metal materials and displacements.

similarity value was used as the positioning result. A confusion matrix was then computed. Although the matrix has a diagonal distribution, significant measurement errors still occur, with an average accuracy of 92.67%. Next, we computed the similarity between the first 100 speckles of each group and the five label speckles, obtaining the similarity curves shown in Fig. 3c.

As the displacement changes, the similarity between the speckles and the reference speckle exhibits a stepwise variation trend. On the one hand, this demonstrates the ability of speckle-based displacement measurement at a 10 nm resolution. On the other hand, the fluctuations in the curve within the same platforms may be caused by environmental micro-vibrations, which increase the measurement error probability of the similarity comparison method.

The speckle similarity comparison method can only handle cases where the target structure remains unchanged, as any change in the structure of the detected object can lead to significant alterations in the speckle pattern, thus affecting displacement estimation. However, in many practical applications, the detected target may change or undergo lateral movement, which may cause the similarity comparison method to fail. In the next experiment, we also selected the same five displacement positions within the 0−40 nm range and simultaneously refreshed 100 different structured patterns on the DMD at each

displacement position. We then calculated the similarity between 500 speckle images and the first speckle, as shown in Fig. 3d. As observed, the results no longer exhibit stepwise variation. The similarity values are closer to forming a random distribution. Under identical patterns, the similarity significantly improves. The periodic structure shown within the red dashed box indicates that the similarity varies according to the periodicity of the pattern structure, rather than the displacement itself.

To address this challenge, we propose a displacement detection learning method based on the VGG convolutional neural network (CNN)[46]. By assigning the same label to different structured patterns under the same displacement and pairing them with the corresponding speckle fields, we trained the network to achieve robustness against variations in the detected patterns. Thereby we establish an intrinsic relationship between the speckle field and the displacement (The network architecture is shown in Methods and Supplementary Note 3 and Supplementary Fig. S5). To demonstrate the system's robustness and repeatability, we performed forward and backward displacement data collection using a nano displacement stage and conducted mixed training. Figure 3e shows the confusion matrix for measurements at a resolution of 100 nm, achieving an accuracy of 99.39%. Further, we collected 20 sets of data at 10 nm displacement

resolution in a single direction, with each set containing 500 speckles, and achieved an accuracy of 98.42%. Figure 3f compares the accuracy for displacement detection under the target's structure changes using the traditional similarity comparison method and the proposed deep learning method, with a 10 nm displacement resolution. Additionally, although neural network-based methods generally require large amounts of data and training costs, our approach achieves an accuracy of 95.48% using just 100 speckles per group as the training set. Figure 3g presents the confusion matrix for measurements at a resolution of l0 nm and 20 groups per 100 datasets. Further experimental results of resolution and range can be found in Supplementary Note 4 and Supplementary Fig. S6–S10.

### Detection of wafers and different metals

In the semiconductor fabrication and damage detection processes, the increasing precision of high-aspect-ratio structures or devices with multiple layers has made it critical to detect nanoscale longitudinal displacements in both the platform and the wafer. To address this challenge, we employed the proposed deep learning-based multimode fiber in situ metrology system to detect wafer displacements. The wafer was mounted on a nano-positioning stage, starting from a position nearly in contact with the fiber-end-ball of the probe, and gradually moved away from the fiber probe in increments of 10 nm. At each displacement position, 500 speckles were captured. Figure 4a presents a microscopic image of the fiber probe at the initial position. With a diameter of only 660 μm, the fiber can access confined spaces in complex systems for detection. The detection accuracy is 99.95%, as shown in the confusion matrix in Fig. 4b. Figure 4c compares the size of the fiber probe with that of a 50-cent Chinese coin, and Supplementary Fig. S22 shows microscopic images of the probe inserted into a quartz capillary tube, demonstrating the probe's ability to operate in confined spaces. In fact, the size of the detection fiber and probe can be further reduced: Probe 1 has a fiber size of 660 μm with a probe diameter of 800 μm, while Probe 2 has a fiber size of 360 μm and a probe diameter of 600 μm.

To accommodate complex pipeline or slit structures inside the equipment, the fiber probe must be able to detect under bending. To address this, we bend a section of the fiber after the fiber probe and altered its bending radius. Displacements at 10 nm resolution were measured for various structured patterns under three bending radii: 5 cm, 7.5 cm, and 10 cm. We demonstrated displacement recognition for each bending radius with individual training. By applying joint learning approach that mixed data from different bending radii under the same displacement condition, it also enabled displacement detection with an accuracy of 99.73% after a single training session across varying bending radii. Figure 4d shows a conceptual diagram of the bending experiment and the results of the joint learning.

On the other hand, due to changes in the scene and the lateral displacement of the target during detection, the material of the detected object may also change. Different materials have distinct reflection characteristics for light fields, which in turn affect the speckles' distribution. To address this issue, we attempted to use the probe for displacement measurement on plates made of different metal materials. Specifically, we conducted forward and backward experiments and sampling on red copper, brass, aluminum, and stainless steel plates at a 100 nm displacement resolution. For the different metals, the multimode fiber system combined with the proposed CNN achieved 100% detection accuracy. Additionally, we employed joint learning to integrate the speckle data from the four different metals. As a result, we achieved not only displacement measurement but also material identification, with an accuracy still reaching 100%. This fully explores the potential of neural networks in optical metrology, enabling multitask processing. Detailed experimental information can be found in the Supplementary Note 7 and

Supplementary Fig. S17. Additionally, it should be noted that the wide-field detection scheme imposes no special requirements on the roughness of the material being tested. As long as the reflectivity is met, the system can be applied to uneven samples. The results and analysis of the roughness measurements for the wafers and metal plates mentioned above can be found in the Fig. S18 of Supplementary Information.

### Speckle sampling and extreme compression ratios

Deep learning frameworks have been widely applied in phase measurement[38], super-resolution imaging[47], and other fundamental scientific research[48,49]. However, this data-driven approach is often criticized for the requirement of large amounts of parallel computing cores and extensive dataset size. Theoretically, displacement measurement does not require the use of full-field speckle patterns for recognition. Therefore, we sampled the speckle field at different regions and sizes, further compressing the sampled data before feeding them into the network. This aims to identify the optimal sampling region and compression ratio for efficient displacement detection.

First, the relationship between sampling location and detection accuracy is explored. Given the circular symmetric degeneracy of transmission modes in MMF, we extract the diagonally symmetric regions from the center to the periphery, at different radii, as shown in Fig. 5a. Next, we select a single diagonal region and compress it to 4×4 pixels. The recognition results for the C1a and C5a sampling regions are shown in Fig. 5b–c, with average accuracies of 91.6% and 74.34%, respectively. The sampling region closer to the center demonstrates a higher information density. In order to achieve small parameter training and increase the training speed, the information compression ratio is further explored. Each point in the output scattering field contains information from all mode components, which effectively contributes to the readout dimensionality reduction. Specifically, we selected the C1a region's sampled speckle, gradually compressing it from 8 × 8 pixels to 2 × 2 pixels, as shown in Fig. 5d. The confusion matrix's probability strength gradually spreads outward from the diagonal, and the accuracy drops from 100% to 62.05%, as shown in Fig. 5e. However, even when the speckles are cropped and compressed to 4 × 4 pixels, which is 0.00315% of all pixels and approximately 0.28‰ of mode number (defined as the information compression ratio), the displacement recognition accuracy still reaches 91.6%. It demonstrates the high performance of the model at extreme compression ratios.

Using high compression ratios speckle fields for displacement recognition helps improve the training speed and reduce hardware configuration requirements. On the other hand, the performance of the system with a small dataset size of 100 speckles per displacement has been shown above. Therefore, to recognize five displacement positions, a standard computational unit like the T4 is sufficient for this small parameter model. In terms of speed, for wafer displacement data, each iteration takes about 2 min. With an A100 computing setup, it is approximately 1 min and 22 s. After 10 epochs, the loss function converges. If the pixel size is compressed to 4 × 4, the total training time (including data preprocessing) can be reduced to 9 min. In terms of data acquisition speed, we significantly increase speckle data collection speed by reducing the region of interest. For instance, within a 128 × 128-pixel region, a standard industrial camera can achieve an acquisition rate of 200 Hz. This means that data acquisition at a single displacement position requires only 2.5 s. For a sparse sampling task involving 20 points, and accounting for the movement time of the displacement stage, the total time required to collect the entire training dataset is less than 1 min. This rapid acquisition speed ensures that even when the target material is not included in the trained dataset, the network can be swiftly corrected.

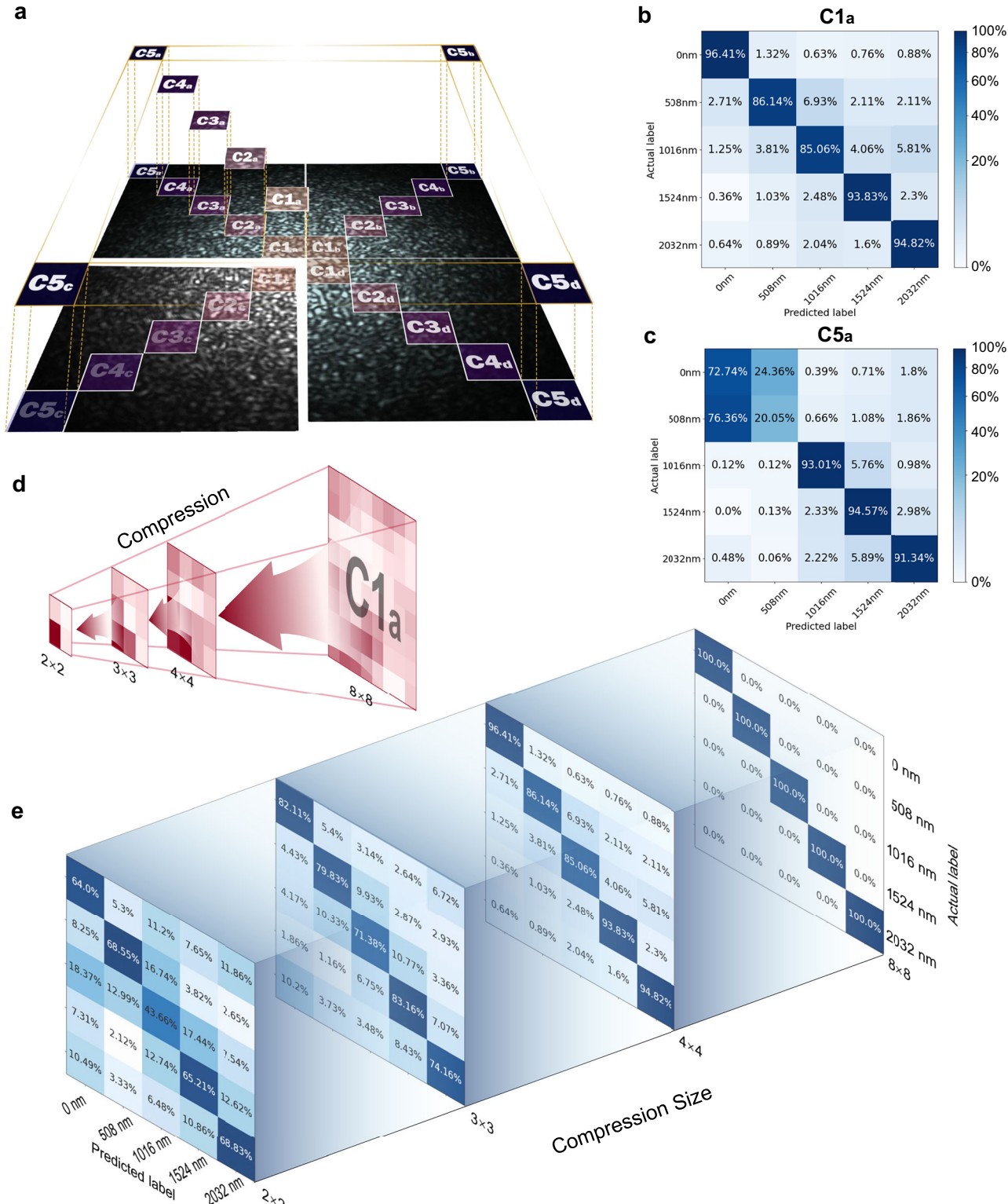

**Fig. 5 | Displacement detection results with compressed sampling speckles.**
**a** Different compressed sampling regions. **b** Displacement confusion matrix of
sampled region C1a compressed to 4 × 4 pixel size at 508 nm resolution.
**c** Displacement confusion matrix of sampled region C5a compressed to 4×4 pixel
size at 508 nm resolution. **d** Compression of the sampled C1a from 8 × 8-pixel to
2 × 2-pixel size. **e** Displacement confusion matrices of the sampled C1a compressed
from 8×8-pixel to 2 × 2-pixel size.

## Image reconstruction with fiber probe

In recent years, multimode fibers have attracted considerable interest
in the field of ultra-thin flexible endoscopy. With their larger infor-
mation capacity and smaller size, they are expected to replace fiber
bundles and provide powerful tools for minimally invasive surgery[50]

and deep brain neuroimaging[25,26,28,29]. As mentioned earlier, the
speckles at the fiber output also contain information about the input
optical field or the target structure. This inverse problem can be solved
by either determining the system's transmission matrix or building
neural networks. Here, image reconstruction is achieved using an ITM

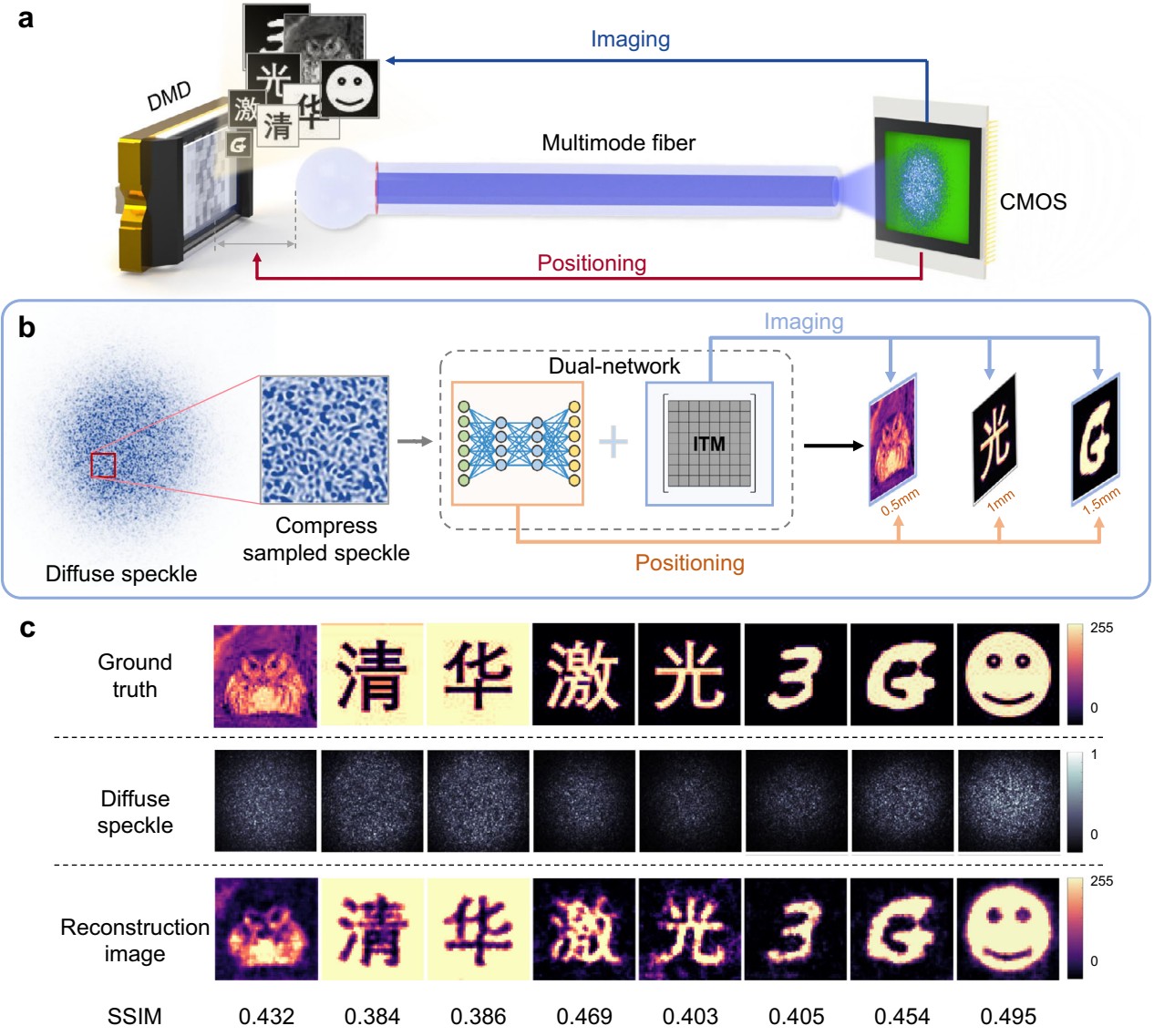

**Fig. 6 | Image reconstruction with the multimode fiber probe. a** Simultaneous positioning and imaging of digital micromirror devices can be achieved with displacement measurements and image reconstruction. **b** The process for image reconstruction and displacement recognition using dual neural networks and compressively sampled speckles. **c** Diffuse speckle field and reconstruction results for grayscale images of natural scenes[63], homemade Chinese characters and patterns.

network. This approach combines the transmission matrix method with a neural network architecture[51]. Compared to solving the full TM, it eliminates the need for a reference optical path and does not require phase information collection. In contrast to traditional deep learning methods, it constrains the neural network through the input-output matrix mapping relationship, resulting in stronger generalization capability. Based on this, we experimentally demonstrated that our multimode fiber detection system can simultaneously perform both positioning and imaging, as shown in Fig. 6a.

Specifically, while collecting displacement data, we refreshed the same set of 4000 grayscale natural scene images on DMD at each position. Simultaneously, CMOS sensors collected the corresponding speckles at the output. The speckles, as shown in the process in Fig. 6b, were first cropped and compressed, then passed through the dual-network (Inverse transmission matrix optimization network and the VGG network). It allowed both displacement recognition and image reconstruction to be achieved. Figure 6c shows the image reconstruction results, which demonstrate that the system can achieve high-quality imaging. The average SSIM between the reconstructed and

detected patterns achieves 0.64. Additionally, after completing network training, it can reconstruct patterns distinct from the training set types. The digits, letters, and icons in Fig. 6c demonstrate test results achieved after training on the grayscale natural scene images, showcasing the network's robust generalization capabilities. We believe that nano-displacement detection fiber probes with imaging capabilities can not only be applied in semiconductor processing and damage detection but also provide precise guidance for minimally invasive surgeries.

## Discussion

In summary, we have implemented an in situ, non-contact nano-displacement measurement system based on multimode fiber's super-oscillatory speckle field and deep learning frameworks. From a physical perspective, we have elucidated the relationship between the displacement of the detected target and the composition of high-order modes in multimode fibers. Compared to traditional similarity comparison methods, our scheme can measure displacements of different structured targets. For wafer displacement, we achieved a resolution

of 10 nm with an accuracy of 99.95%. The system's adaptability to different bending radii has been validated, demonstrating its capability to extract displacement information even in deeply curved narrow slits. By joint learning, we achieved both displacement and material recognition for four different types of metal materials. The dataset size can be reduced to 100 speckles per group. Readout dimensionality can be reduced to 0.28‰, still achieving an accuracy of 91.6%. With a dual-network structure, the system can simultaneously perform both displacement measurement and image reconstruction tasks.

In terms of measurement range, we have validated the system's detection ability within a 2 mm range. We performed displacement recognition at 0.5 mm intervals, covering a range from 0-2 mm (see Supplementary Figs. S8 and S28). However, due to computational constraints, we carried out recognition for 100 displacement groups at 10 nm resolution (Average accuracy of 97.8%, see Fig. S9 in the Supplementary Information). With more advanced computational units and larger memory capacity, the system can achieve a larger measurement range at nano-resolution. Regarding measurement resolution, due to limitations of the displacement stage, our system has currently demonstrated single-direction accuracy of 10 nm and return accuracy of 100 nm. However, we believe that by applying a higher-precision displacement stage, we can further explore the minimum resolution of the multimode fiber displacement measurement system. Although the preceding discussion only demonstrated discrete displacement recognition results obtained using classified-like networks, neural networks based on superoscillatory speckle can output continuous displacement values through strategies such as Gaussian Label Regression and encoded rectangular patterns. Detailed methods and experimental results are presented in Supplementary Note 5 and Supplementary Figs. S11–S14.

To achieve flexible detection and portability, our system is highly integrated, with no discrete lens components in the entire optical path. The objective lens at the detection end is replaced by a fiber-end-ball, and the output end face is directly projected onto the CMOS surface. This enables a all-fiber-based and lens-free system. Furthermore, we have observed that the diffuse speckle field offers superior performance in displacement measurement compared to the focused speckle field produced by an objective (see Supplementary Fig. S25). This could be because the diffuse speckle field amplifies the variations between higher-order modes, which aligns with the conclusions from the physical model discussed in the results section.

In addition, we have integrated the inverse transmission matrix network with the VGG displacement recognition network to extract the ITMs for different displacements. These ITMs are labeled with displacements and stored in a displacement-inverse transmission matrix library (DITML). During actual detection, the displacement is first recognized, and then the most suitable ITM is searched from DITML. This matrix is then used for image reconstruction. By employing this approach, we can effectively enhance imaging quality compared to the joint learning method[52]. Detailed implementation steps are provided in Supplementary Fig. S27 and Supplementary Movie 2.

Furthermore, we have combined displacement measurement and image reconstruction functions to preliminarily achieve the reconstruction of three-dimensional micro-nano structure profiles. The system not only recognizes the displacements but also superimposes three-dimensional structural cross-sections at different displacements, enabling the reconstruction of three-dimensional surface profiles (see Supplementary Note 12 and Supplementary Figs. S28 and S29).

The nano-positioning is fundamental to lithography[53] and is crucial for wafer positioning, mask alignment, and semiconductor detection. Additionally, fields such as molecular biology[54], particle tracking[14], nanoassembly, and nanofabrication[55] are also in dire need of high-performance micro-imaging and positioning technologies. The proposed in situ non-contact small-size fiber probe for longitudinal displacement measurement can be applied to nano-positioning systems[56] without the need for a vacuum environment. In future studies, it is expected that multimode fiber ruler and hypersurface technology can be used to combine longitudinal displacement measurement with lateral displacement measurement, which will enable the reconstruction of micro-three-dimensional surface profiles. On the other hand, by addressing the challenges associated with preparing datasets for real-world targets, the potential for three-dimensional imaging beyond the diffraction limit could be realized. Additionally, the system can incorporate new degrees of freedom, such as spectral information, which also influences modal dispersion and inter-mode coupling. The speckle field can restore spectral information while potentially further enhancing displacement recognition accuracy. With the physical properties of flexible and small diameter, lensless integrated MMF micro-detection system paves the way for portable endoscopes[57,58], and potential applications in implantable devices, such as brain observation in active mice[59,60]. At the same time, we believe that the optical super-oscillatory speckle field observed in multimode fibers may also exist in complex media with similar scattering properties[61,62]. Therefore, the proposed approach holds potential for further expansion and application in these areas.

## Methods

### Physical simulation model

In the model for the relationship between detected displacement and higher-order mode composition in multimode fiber, we apply the weak guidance approximation. Using mode decomposition, we decompose the input optical field into different transverse modes of step-index MMF. We then use angular spectrum methods to analyze the optical field. After it is reflected from the detected target, passing through free-space transmission and the fiber-end-ball transformation, reaching the input facet of the fiber. Here, we treat the fiber-end-ball as a spherical lens with a fixed radius. The detailed calculation process can be found in the Supplementary Note 1. Based on the orthogonality of the linearly polarized modes, we use the cosine projection method to compute the projection coefficients from the Cartesian coordinate basis to the LP mode coordinates,

$$C_{i(m,n)} = \frac{\int_0^\infty \int_0^\infty LP_{mn}^*(x,y) U_4(x,y) dx dy}{\int_0^\infty \int_0^\infty LP_{mn}^*(x,y) LP_{mn}(x,y) dx dy} \quad (2)$$

Where $U_4(x,y)$ is the light field at the input end face of the fiber, $LP_{mn}(x,y)$ represents the distribution of linearly polarized mode fields with angular mode order and radial mode order of m and n, respectively. $i$ represents the total mode order, which is obtained by arranging these LP mode fields in the order of the cutoff frequency. In this experiment, the first 1000 modes are taken into consideration. Project the input mode field to the first 1000 LP mode fields to get the mode coefficients $C_1 \sim C_{1000}$. In order to more intuitively evaluate the energy share of the input light field decomposed into each mode, further calculations were performed to obtain the energy percentage factor:

$$P_I(i) = \frac{|C_i|^2 \int_0^\infty \int_0^\infty LP_{mn}^*(x,y) LP_{mn}(x,y) dx dy}{I_{U_4}} \quad (3)$$

Where, $I_{U_4}$ is the total input energy of $U_4$,

$$I_{U_4} = \int_0^\infty \int_0^\infty U_4^*(x,y) U_4(x,y) dx dy \quad (4)$$

### Experimental set-up

The laser beam of a 1064 nm wavelength laser (Connet, CoSF-D-YB-M-1064-PM-FA) with a narrow linewidth (20 kHz) is used to illuminate. The MMF used to detect the displacement is a double cladding step-

index multimode fiber with 250 µm diameter and numerical aperture (NA) of 0.062(Nufern, LMA-GDF-30/250-M). The distal end of the double-clad fiber is fused to a triple-cladding multimode fiber (Nufern, BD-S200/230/660-STN) to increase the field of view. The distal facet of the triple-clad multimode fiber is melted into a fiber-end-ball with 800 um diameter, which homogenizes and collimates the output beam. The illumination light wave detects the displacements of the wafer(S-MIC), digital micromirror device (DMD, Texas Instruments DLP LightCrafter 4500), and metal plates. Then, it carries the information back into the MMF. The amount of displacement is precisely controlled by a nanoscale electric displacement stage (COREMORROW, P18.X300S & New focus, Model 8751-C). The signal light transfers through the MMF and then emerges from the facet, forming speckle patterns which are received by a sensor (Sony, 1/1.8" IMX252 Global shutter CMOS) from the camera (Daheng imaging MER2-301-125U3M). The specific structure of the measurement system is described in the Supplementary Fig. S21.

### Fiber devices fabrication

Side couplers for integrating illumination fibers and detection fibers are fabricated on a fiber optic fusion and taper drawing platform. The fabrication process is mainly divided into two steps. First, one end of an illumination fiber (YOFC, 200/220 µm, 0.22/0.46 NA) is drawn into a tapered fiber using a hydroxide flame taper machine. The length of the tapered area is 2 cm. In the next step, the tapered area of the illumination fiber is affixed to the detection MMF, by fusing and pressing it together.

Instead of conventional discrete lenses, fiber-end-ball at the distal end of the probe is used to homogenize and collimate the illumination light and to couple the signal light into the MMF. We use a $CO_2$ laser processing and splicing platform for the preparation of fiber-end-ball. One end of the stripped-coated triple-clad MMF is cut flat using a large diameter optical fiber cleaver (Vytran, LDC-400). After that, it is pushed into a heating area at a certain speed (0.03 mm/s) and uniformly heated by two $CO_2$ laser beams. While being heated, the MMF is rotated at a set speed (50 °/s) to further ensure uniform heating. The fiber in the heated area is thermally expanded and the high-speed rotation gives good symmetry to the prepared end-balls.

The receiving end of the double-cladding MMF is stripped of the coating layer and then cut flat by the large diameter optical fiber cleaver. It is integrated with the image sensor equipped on the displacement stage to realize lensless output.

### Dataset preparation

In the wafer displacement detection and metal material detection experiments, we collected 10 sets of data, each consisting of 500 speckle images, in the range of 0–400 nm with a resolution of 10 nm or 100 nm for both forward and backward measurements. After cropping and compressing the speckles, we input them into the VGG neural network. In the displacement detection and image reconstruction experiments, processed square natural scene grayscale images and self-made 3D structural cross-section views are used as the dataset. 4000 sets of images and corresponding diffuse speckle fields are collected at each measuring distance. Among them, 70% are used as the training set, 10% as the validation set, and 20% as the test set. Due to the limitation of the DMD pixel size and field of view, we set the pixels of the ground truth to 48 × 48. Since the diffuse speckle field does not have an obvious boundary, each image recorded by the CCD is cropped by a fixed size. And the scattering center was marked by a certain intensity threshold. Then they are compressed to 128 × 128 pixels as input to the network.

### Neural network architecture

The diffuse speckle field, compressed into a 128 × 128 pixels resolution, is initially processed by the VGG displacement recognition network.

The input first traverses a 3 × 3 convolutional layer, followed by a hyperbolic tangent (tanh) activation function and a 2 × 2 max-pooling layer. This sequence is repeated iteratively, with the channel depth incrementally doubled from 32 up to 256. Throughout these iterations, the kernel size, activation function, and pooling parameters remain constant. The resulting output is subsequently flattened and input into a fully connected (FC) layer of dimensions 1 × 16384, where the tanh activation function is applied. Following this, two additional FC layers sequentially reduce the dimensionality from 16384 to the number of displacement samples (N).

For the inverse transmission matrix optimization network, the compressed speckle is first complexified, as the transmission matrix is a complex matrix. It is then fed into the complex dense layer. Here, an initialized and normalized complex inverse transmission matrix is multiplied with the input diffuse speckle. Since the partial speckle distribution also contains all the information of the input image, the reconstructed image can be obtained with a sufficient amount of information. The output is then compared with the ground truth and the inverse transfer matrix is further optimized using a neural network feedback framework. The final output are reconstructed images after convergence of the loss function.

By combining the two networks, displacement detection and imaging quality optimization are achieved. We utilize the colab computing platform provided by Google for network training, deployed using T4 and A100 computing units. We use the structural similarity parameter for evaluation in both the speckle similarity comparison method as well as image reconstruction,

$$SSIM(y, y_R) = \frac{\left(2\mu_y\mu_{yR} + c_1\right)\left(2\sigma_{yyR} + c_2\right)}{\left(\mu_y^2 + \mu_{yR}^2 + c_1\right)\left(\sigma_y^2 + \sigma_{yR}^2 + + c_2\right)} \tag{5}$$

where $\mu_y$, $\mu_{yR}$ are the means of the real image y and the recovered image $y_R$, respectively. $\sigma_y^2$, $\sigma_{yR}^2$ are the variances of $y$ and $y_R$, respectively. And $\sigma_{yyR}$ is the covariance of y and $y_R$. $c_1 = [k_1(2^{bits}\text{-}1)]^2$, $c_2 = [k_2(2^{bits}\text{-}1)]^2$ are constants used to maintain stability and avoid divisors to be 0. $k_1 = 0.01$, $k_2 = 0.03$. The closer the SSIM value is to 1, the more similar the two images are and the higher the image reconstruction quality is.

Specific network architecture diagrams and parameter configurations are given in Supplementary Note 3 and Supplementary Fig. S5.

## Data availability

All datasets generated or analysed during this study are described in detail in the main text and the Supplementary Information. Due to the large volume of the raw data, they are not deposited in a public repository. The datasets are available from the corresponding authors upon reasonable request.

## Code availability

The code used for training the neural network is available at https://doi.org/10.5281/zenodo.17525001.

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

## Acknowledgements
We acknowledge funding support from the National Natural Science Foundation of China (62475132, 62122040, Q.X.), Beijing Natural Science Foundation (L241021, Q.X.).

## Author contributions

L.W. conceived the experimental system. L.W. and Y.Zhang performed the experiments. Y.Zhang and Y.Zhou developed the algorithm. L.W., Y.Zhang, Y.Zhou and Z.L. processed and analyzed results. P.L fabricated fiber devices. P.Y., D.L. and Q.L directed the research. L.W. and Y.Zhang prepared the manuscript. Y.M. and H.Z. polished the manuscript. All authors contributed to the writing of the paper. Q.X. and Q.L conceived and led the project.

## Competing interests
The authors declare no competing interests.
