## [Transparent Peer Review file · Nature Communications]

Deep learning and superoscillatory speckles empowered multimode fiber probe for in situ nano-displacement detection and micro-imaging

Corresponding Author: Professor Qirong Xiao

Version 0:

Reviewer comments:

Reviewer #1

(Remarks to the Author)

This manuscript entitled “Deep learning and superoscillatory speckles empowered multimode fiber (MMF) probe for in situ nano-displacement detection and micro-imaging” by the authors proposed an interesting and useful methodology of in situ nano-displacement detection and imaging based on the combination of super-oscillatory speckles of MMF and deep learning, demonstrating single-ended displacement detection with 10 nm resolution and endoscopy capability. By using a physical model linking displacement dependent diffraction patterns of the object under detection and their corresponding energy proportion of higher-order modes, the authors reveal the mechanism to achieve nanoscale displacement sensing and imaging using the speckle fields. Robust performance under different bending conditions is also demonstrated, enabling flexible deployment in pipeline environments mediated by the all-fiber device. Before the recommendation of publication for this manuscript, the following issues should be fully addressed:

1. The definitions of “fiber input cross-section” and “optical field at the core area” mentioned in the caption of Fig. 2 b-e are confused. In the caption, “optical field at the core area” is used while “input optical field” is marked in Fig. 2d. It is also confused for this sentence in the main text: “Fig. 2c and 2d show the fiber detection end facet and input optical field distribution after free space transmission and fiber-end-ball transformation, respectively.” And there is no “fiber input facet” indicated in Fig. S1. The description in Supplementary Note 1 is confused as well. Because the definitions of planes along the propagation direction under study in the Supplementary text are inconsistent with the main text. The authors should unify the definitions because this is a key point to interpret the sensing and imaging mechanisms using modal decomposition.
2. Following previous comment, if my understanding is correct, the authors perform modal decomposition at the U4 plane shown in Fig. S1. At this plane, whether the modelling of diffraction in uniform material is appropriate? Because there are boundaries imposed by the core and cladding.
3. The linear polarization mode basis is used for the modal decomposition. Why the authors choose this basis? The linear polarization modes are not the eigenmode of an MMF.
4. Fig. 2f shows the excited LP mode orders when Z ranges from 5 mm to 35 mm. Could the authors provide data on LP modes within the 0–300 nm displacement range, as the most important displacement sensing demos in the study are within this range?
5. The authors demonstrated the robustness of the displacement recognition network under various fiber bending conditions. But the authors did not evaluate whether the inverse transmission matrix imaging network can maintain image reconstruction quality under various fiber bending conditions.
6. There is no discussion for Fig. 2g. And why there are different fluctuation situations in the excited modal orders of Fig. 2f and 2g.
7. A detailed description of the inverse transmission matrix optimization network should be provided. Or relevant reference to it should be provided, if any.
8. Whether the sequential order of the dual-network in Fig. 6b is correct? As discussed by the authors, whether the neural network should sense the displacement first and then find the corresponding ITM in the displacement dependent ITM library.

(Remarks on code availability)

Reviewer #2

(Remarks to the Author)

This manuscript presents an in situ displacement measurement technology based on a multimode fiber probe generating superoscillatory speckles. Combined with a VGG convolutional neural network, the system achieves non-contact detection at 10 nm resolution and 99.95% accuracy, demonstrating robust performance under varying fiber bending radii and materials via joint learning. The lens-free imaging capability of the system is also experimentally validated. However, these results are insufficient to guarantee publication in Nature Communications for several reasons. First, multimode fiber probes have been vastly studied for endoscopic imaging, and extending their application to distance measurement here involves key technologies already widely discussed in endoscopy, lacking sufficient technical originality. Second, for displacement measurement (especially longitudinal displacement), white-light interferometry-based techniques already achieve sub-nanometer precision, making the reported 10 nm accuracy in this work technically unadvanced. Additionally, the scheme requires object-specific training, limiting its generality as a universal device.

Additional concerns:

- (1) As shown in Figure S4a, the VGG displacement recognition network is a classification network with an output layer of five neurons corresponding to 0 nm, 10 nm, 20 nm, 30 nm, and 40 nm. This method can only predict discrete displacement values (10 nm resolution). To achieve continuous displacement measurement with higher resolution, is it feasible to develop a neural network that directly outputs specific displacement values?
- (2) In Figure S18, the 2 mm measurement range is validated at 0.5 mm intervals, which are excessively large. Simultaneously achieving high resolution and long measurement range would require a huge, complex network (e.g., 200,000 neurons in the output layer for 10 nm resolution over 2 mm). Moreover, the volume of speckle data requiring calibration across different displacements is enormous. Are there effective strategies to balance long range and high resolution?
- (3) In Figure 4f, displacement measurement and material identification achieve 100% accuracy at 100 nm resolution, but the intervals are large. Will increasing the number of neurons in the output layer (for higher resolution or longer range) lead to a decrease in accuracy?
- (4) The influence of the distance between the speckle detection plane (CMOS sensor) and the fiber output face should be analyzed.
- (5) The system trains a dual network on 4,000 images. After training, can this network accurately detect displacements and reconstruct images for objects beyond these 4,000 images?
- (6) The scheme's general applicability is limited, as detecting arbitrary objects would require collecting vast amounts of speckle data for different materials.
- (7) The triangular and square markers in Figure 2h are mislabeled.

(Remarks on code availability)

Reviewer #3

(Remarks to the Author)

The authors present a non-contact longitudinal nano-displacement metrology system utilizing multimode fibers (MMF) and deep learning framework. Key innovations include: 1) a physical model correlating input images, MMF mode compositions, and output speckle patterns; 2) multitask processing validity for simultaneous displacement measurement (with 99.95% accuracy and 10 nm resolution) and target imaging; 3) a compact and highly integrable lensless single-end MMF detection system with applications in semiconductor industry and super-resolution endoscopy.

While the manuscript demonstrates compelling results, the following issues require clarification before recommending publication in Nature Communications:

Major issues:

1. How would reduced fiber probe diameters (for example to use fiber probe 2 in Fig. 4c) affect the measurement accuracy?
2. It will be more fundamental to study the dependence of displacement measurement on the surface roughness. If the roughness is comparable or even above the minimum resolution (10 nm for example), is the deep learning algorithm still applicable and reliable?
3. There is a technical challenge that the fiber must be aligned vertically with respect to the sample surface, otherwise the fields of view will change at different heights. If the fiber is slightly tilted, what will happen in the output speckle fields? How about tolerance on the misalignment angles?
4. The proposed scheme requires a large amount of data for training purposes, and the whole process is quite time-consuming (to take 500 speckle fields per step in several minutes), making it susceptible to environmental vibrations. Any comments on how to balance the accuracy and time?
5. Speckle field is sensitive to the laser coherence. What's the linewidth and coherence length of the laser source? Is it possible to use the spectrum or wavelength as a new degree of freedom for displacement measurement? A brief discussion will be beneficial.
6. In the section of "Image reconstruction with fiber probe", if the input image is not in the ImageNet dataset, is it still possible to recognize it using the image reconstruction scheme? If yes, add an experimental result in Fig. 6c.
7. In Supplementary Note 4, it's mentioned that "Laser powers also need to be adjusted, that the maximum grey value at the

center of the speckle is consistent at different positions.” Is the laser power adjusted manually or automatically? If this step is mandatory, the calibration process will be very time consuming. For a large displacement range up to a millimeter level, the intensity will change dramatically, how to make sure the calibration reasonable? Will the measurement accuracy be improved if using a large dynamic range CCD, for example 10 bit?

8. Throughout the manuscript, the authors mainly focused on the longitudinal displacement metrology. How about the transverse resolution? Any qualitative prior knowledge is expected. What could be the ultimate longitudinal and lateral resolution and limiting factors?

Minor issues:

9. In the description to Fig. 6c, the SSIM is claimed to be 0.64. Here it should not be a single value but a range. Or simply use ‘averaged SSIM’ for clarity? If the authors didn’t crop and compress the input images, will SSIM be increased?

10. In Fig. 2f row 1, check if the displacement 0.5 mm or 5 mm. The data in the figure, figure caption and main text are not consistent.

11. The U_1 in equation(S2) should be U_{1F} .

12. H_{AS} should use a different quantity name in (S8) to differentiate from its counterpart in (S3).

13. In Fig. S13, the labeled ball lens diameter should be 800 μm according to the information provided.

(Remarks on code availability)

Version 1:

Reviewer comments:

Reviewer #1

(Remarks to the Author)

The authors have fully addressed all my comments and I am willing to recommend the publication of this manuscript in Nature Communications.

(Remarks on code availability)

The codes are available and useful for reproducing the results of the manuscript.

Reviewer #2

(Remarks to the Author)

The authors have answered my questions and the manuscript is acceptable now.

(Remarks on code availability)

Reviewer #3

(Remarks to the Author)

The authors have comprehensively and thoughtfully addressed all my questions and revised the manuscript accordingly. Specifically, they investigated how the accuracy of longitudinal displacement measurement depends on key parameters: fiber probe size, surface roughness, fiber alignment angle, and laser coherence. I am pleased to recommend this work for acceptance in Nature Communications in its current form.

(Remarks on code availability)

I downloaded the code; however, as I am not personally familiar with Python, I am unable to provide a fair assessment of it.

Author Responses

We firstly would like to sincerely thank the reviewers for the constructive comments. The suggestions were well taken.

In accordance with these suggestions, we have conducted extensive experiments and simulation analyses on the multimode fiber displacement metrology system. Both our manuscript and supplementary materials have been carefully revised following the reviewers' instructions. Below, we provide point-by-point responses to the comments. Each comment is listed first, followed by our corresponding reply.

In brief summary, based on the reviewers' constructive feedback and suggestions, we have made the following revisions to the manuscript:

1. Enhanced simulations of optical field evolution and mode decomposition in the multimode fiber. The example target has been replaced with a wafer micrograph. Additionally, the relationship between mode composition and displacement has been supplemented under small displacements (0–1000 nm) at high resolution.
2. Extended displacement recognition experiments to a larger range (0–990 nm) with 100 sampling points.
3. Thoroughly discussed potential improvements for the displacement recognition network, proposing two feasible strategies: Gaussian Label Regression and encoded rectangular patterns.
4. Included an introduction to the inverse transmission matrix optimization network, with experimental validation of the generalization ability.
5. Added experimental details regarding system data acquisition time, light source linewidth, and target roughness, demonstrating the system's capability for rapid calibration and training.
6. Standardized some definitions in simulations and refined details in Fig. 2 and 6.

In addition, we have included further experimental details and results in the Supplementary Materials, such as: multi-interval partitioned recognition strategy that enables both large dynamic range and high resolution; imaging experiments under different bending states; roughness characterization of test samples using atomic force microscopy; displacement recognition capability under various acquisition distances and detection angles.

We wish to extend our sincere appreciation once again to the reviewers for the time and effort dedicated to evaluating our manuscript and for providing insightful feedback. We are confident that the revised version demonstrates substantial improvements in both systematic performance and overall completeness, thereby more effectively underscoring the novelty and strengths of our work.

Contents

RESPONSE TO REVIEWER 1	3
COMMENT 1	3
COMMENT 2	5
COMMENT 3	7
COMMENT 4	9
COMMENT 5	15
COMMENT 6	19
COMMENT 7	20
COMMENT 8	25
RESPONSE TO REVIEWER 2	26
COMMENT 1	30
COMMENT 2	35
COMMENT 3	40
COMMENT 4	42
COMMENT 5	44
COMMENT 6	44
COMMENT 7	47
RESPONSE TO REVIEWER 3	48
MAJOR ISSUES:	48
COMMENT 1	48
COMMENT 2	52
COMMENT 3	55
COMMENT 4	59
COMMENT 5	64
COMMENT 6	65
COMMENT 7	66
COMMENT 8	67
MINOR ISSUES:	69
COMMENT 9	69
COMMENT 10	70
COMMENT 11	71
COMMENT 12	71
COMMENT 13	72

Response to Reviewer 1

This manuscript entitled “Deep learning and superoscillatory speckles empowered multimode fiber (MMF) probe for in situ nano-displacement detection and micro-imaging” by the authors proposed an interesting and useful methodology of in situ nano-displacement detection and imaging based on the combination of super-oscillatory speckles of MMF and deep learning, demonstrating single-ended displacement detection with 10 nm resolution and endoscopy capability. By using a physical model linking displacement dependent diffraction patterns of the object under detection and their corresponding energy proportion of higher-order modes, the authors reveal the mechanism to achieve nanoscale displacement sensing and imaging using the speckle fields. Robust performance under different bending conditions is also demonstrated, enabling flexible deployment in pipeline environments mediated by the all-fiber device. Before the recommendation of publication for this manuscript, the following issues should be fully addressed:

Comment 1: The definitions of “fiber input cross-section” and “optical field at the core area” mentioned in the caption of Fig. 2 b-e are confused. In the caption, “optical field at the core area” is used while “input optical field” is marked in Fig. 2d. It is also confused for this sentence in the main text: “Fig. 2c and 2d show the fiber detection end facet and input optical field distribution after free space transmission and fiber-end-ball transformation, respectively.” And there is no “fiber input facet” indicated in Fig. S1. The description in Supplementary Note 1 is confused as well. Because the definitions of planes along the propagation direction under study in the Supplementary text are inconsistent with the main text. The authors should unify the definitions because this is a key point to interpret the sensing and imaging mechanisms using modal decomposition.

Response:

We sincerely appreciate your recognition of our work and your insightful suggestion regarding the definition of the optical field at different locations within the simulation. This point was indeed prone to misinterpretation, as it inherently involves two distinct physical models for optical field simulation: the Free-Space-to-Fiber Coupling Model utilizing angular spectrum diffraction, and the Mode Decomposition and Recomposition Model based on mode decomposition. Within the latter model, the "**input optical field**" intrinsically refers to the "**optical field at the core area**" after coupling. This has indeed led to two distinct definitions. Accordingly, we accept your suggestion and have uniformly defined this field as the "**Field at the core area**" throughout Fig. 2, its caption, and the main text. The revised figure and corresponding text section are presented below.

(From page 4, line 149)

and low-frequency structures, to excite higher-order modes and obtain universal results. Fig. 2c and
2d show the fiber input facet and field at the core area after free space transmission and fiber-end-
ball transformation, respectively. We performed mode decomposition on the input optical field and

(From page 5, line 165)

Owing to space constraints in the main text, the Free-Space-to-Fiber Coupling Model was not detailed therein. This decision was made because the results from the Mode Decomposition Model are more directly pertinent to the subsequent validation of high-precision positioning capability, as addressed later in the manuscript. As you correctly noted, the Supplementary Information present the Free-Space-to-Fiber Coupling Model. Crucially, the optical field plane defined within this supplementary model differs from that defined in the main text, as they belong to distinct simulation frameworks.

To prevent potential confusion, we have added the optical field planes defined in the main text—namely, the "Field at the core area" and the "First 1000 mode combination"—to the Supplementary Information. Furthermore, the definition of U4 has been revised to "Fiber input facet", ensuring consistency with the terminology used in the main text. The figure also adds annotations for the Free-Space-to-Fiber Coupling Model and the Mode Decomposition and Recomposition Model. The specific modifications are illustrated in Figure S1. The revised figure from the Supplementary Information is presented below.

(Supplementary Information, from page 23, line 917)

[editorial note: third party material redacted]

Comment 2: Following previous comment, if my understanding is correct, the authors perform modal decomposition at the U4 plane shown in Fig. S1. At this plane, whether the modelling of diffraction in uniform material is appropriate? Because there are boundaries imposed by the core and cladding.

Response:

Thank you for your reminder and suggestion. As you understood, we performed modal decomposition on the optical field at the U4 plane to analyze the impact of target displacement changes on the mode proportion within the fiber's optical field. However, we did not use a diffraction model of homogeneous material for the modal decomposition. In the calculation of front-end free-space optical transmission and fiber-end-ball transformation, we utilized the angular spectrum diffraction model.

In the modal decomposition part, as you mentioned, the multimode fiber has two materials with different refractive indices: the core and the cladding. Boundary conditions with abrupt refractive index changes exist between the core and cladding, and between the cladding and air.

In fact, during the calculation of LP modes, we incorporated these considerations. This boundary condition was used to constrain the optical field propagation. The first 1000 transverse modes were all simulated and calculated under the condition of an inhomogeneous waveguide with step-index, as shown in Fig. R1a.

Moreover, when coupling the front-end optical field to U4, we selected the entire optical field within the fiber for modal decomposition, rather than only the optical field within the core. Therefore, we included the leaky optical field in the cladding into the modal decomposition model, as shown in Fig. R1b. Based on this, we were able to obtain a mode combination with extremely high reconstruction similarity and an energy utilization rate exceeding 95% after decomposition. This is more conducive to the subsequent analysis of displacement impact.

Fig. R1 LP mode distribution in the core and cladding regions and optical field distribution within the multimode fiber. a, Examples of low-order LP mode distributions calculated under the boundary condition introducing the refractive index step discontinuity between the core and cladding. **b,** Optical field distribution on the fiber input facet before mode decomposition, including the leaky optical fields within the cladding.

Descriptions in the text may have contained potentially confusing elements. We have revised the relevant descriptions in the main text to emphasize the concept of performing modal decomposition in a step-index inhomogeneous medium, as follows:

(From page 4, line 140)

displacements. A MMF physical model is developed to simulate the propagation and coupling
process of the returned signal light from the detected target by calculating the input optical field
evolution of step-index MMF and mode decomposition⁴². The mode decomposition coefficients of

(From page 14, line 478)

**Methods**[↵]
**Physical simulation model**[↵]
In the model for the relationship between detected displacement and higher-order mode composition
in multimode fiber, we apply the weak guidance approximation. Using mode decomposition, we
decompose the input optical field into different transverse modes of step-index MMF. We then use

Comment 3: The linear polarization mode basis is used for the modal decomposition. Why the authors choose this basis? The linear polarization modes are not the eigenmode of an MMF.

Response:

Thank you for your question. In our modal decomposition model, we used linearly polarized (LP) modes as the basis for decomposition. This is because the relative refractive index difference between the core and cladding in most multimode fibers satisfies the weak-guidance approximation.

$$\Delta = \frac{n_1^2 - n_2^2}{2n_1^2} \approx \frac{n_1 - n_2}{n_1} \approx \frac{n_1 - n_2}{n_2} \ll 1 \quad (1 - 1)$$

Where n_1 and n_2 are the refractive index of the fiber core and cladding, respectively. Under this approximation, the transverse electromagnetic field dominates, and the polarization state of the optical field remains unchanged during propagation along the fiber. Additionally, the evolution of the input optical field along the multimode fiber can be simply described by transforming a phase parameter without altering the spatial field distribution. Therefore, these LP modes are also referred to as propagation-invariant modes (PIMs).

As shown in Fig. R2a, the transmission matrix of the multimode fiber acquired under the spatial orthogonal basis is a complex, irregular large matrix. When transformed into the LP mode basis, the transmission matrix converts into the form shown in Fig. R2b. This converted matrix exhibits distinct diagonal characteristics, demonstrating the transmission-invariant property of LP modes (Ref: Nature Photonics, 9, 529-535, 2015).

[editorial note: third party material redacted]

Fig. R2 Transmission matrix of a 10mm-long multimode fiber under different bases (Ref: Nature Photonics, 9, 529-535, 2015). **a**, Transmission matrix measured in the spatial orthogonal coordinate system. **b**, Transmission matrix under LP PIMs after coordinate system transformation, exhibiting clearly visible diagonal characteristics.

Using spatial orthogonal bases or vector mode decomposition would make it difficult to intuitively establish the relationship between the input optical field and the output speckle field. This would prevent the evaluation of displacement measurement effects and complicate the model. On the other hand, in the field of multimode fiber imaging research, most studies also employ propagation-invariant modes for decomposition. We believe this approach facilitates wider recognition and impact within related research domains.

Certainly, under the scalar approximation, fiber modes exhibit degeneracy. Moreover, due to polarization coupling, LP modes cannot be considered propagation-invariant in long-distance fibers. Further utilizing the weak-guidance approximation provides a more accurate description than the scalar approximation. It considers the mutual influence between the polarization state of light and spatial field distribution, known as spin-orbit (SO) interaction.

After introducing SO interaction, circular polarization bases $\hat{\sigma}_{\pm} = (\hat{x} \pm i\hat{y})/\sqrt{2}$ can replace the linear polarization bases \hat{x} and \hat{y} . The resulting propagation-invariant modes can then be termed circular polarization propagation-invariant modes (CP PIMs). However, since

our study focuses on investigating the influence of target displacement on multimode fiber output speckle fields and mode composition—rather than the field evolution between input/output (which the fiber imaging experiments focus on more)—we maintain our approach.

In fact, we contend that any basis satisfying orthogonality and completeness can serve as the decomposition basis and be regarded as characteristic modes of multimode fiber. Therefore, we selected the more widely adopted linear polarization modes in this field as the basis for modal decomposition in our paper.

Comment 4: Fig. 2f shows the excited LP mode orders when Z ranges from 5 mm to 35 mm. Could the authors provide data on LP modes within the 0–300 nm displacement range, as the most important displacement sensing demos in the study are within this range?

Response:

We thank you for this insightful suggestion. We fully agree that it is important to investigate the LP mode behavior in the displacement range of 0–300 nm, which is indeed relevant to the main experimental examples in our study.

To address this comment, we focused on the 0–1 μm displacement range, which corresponds to the full range (0–990 nm) used in our displacement recognition task at 10 nm resolution. In order to better align with the application context of our research and enhance the overall integrity of the manuscript, we replaced the Lenna image with a wafer surface micrograph.

As can be seen from Fig. R3 below, as the displacement distance increases, the normalized average mode order shows a monotonically increasing trend, while the proportions of both the first 100 LP modes and the first 1000 LP modes demonstrate monotonically decreasing trends. This finding contradicts the laws observed in the large-range measurements of the original manuscript. We speculate that this discrepancy may be attributed to the detection target not reaching the equivalent focal plane of the fiber-end-ball. To verify the reliability of these patterns, we compared inputs with different wafer patterns. The output results showed a high degree of consistency.

Fig. R3 Variation of normalized mode order, percentage sum of first 100 modes, and percentage sum of first 1000 modes with displacement distance under small-range high-resolution measurement (0–990 nm).

To further investigate the patterns under large displacement ranges, we have supplemented the simulation results of wafer microscopic images within a large displacement measurement range of 0–10 mm in the Supplementary Information. Sampling points were taken at 1 mm intervals, and scatter plots of the normalized mode order, the percentage sum of first 100 LP modes, and percentage sum of first 1000 LP modes were plotted along with fitted curves, as shown in Fig. R4.

It can be observed that, although fluctuations exist in the normalized mode order and the percentage sum of first 100 and first 1000 LP modes at certain positions, an overall monotonic

trend is evident. Within the large displacement range, the proportions of both the first 100 and the first 1000 LP modes exhibit an increasing trend, while the normalized average mode order shows a decreasing trend. These results are consistent with the conclusions drawn in the original manuscript. It is observed that the variation trends at short and long displacement ranges are opposite. We speculate that the reason why the normalized mode number first increases and then decreases is that, as the displacement increases, the system transitions from a defocused state to focus and then back to defocus. Since defocus leads to a reduction of high-frequency components (Ref: *Proc. R. Soc. London, Ser. A*, 231, 91–103, 1955), the high-frequency components of the light field coupled into the fiber first increase and then decrease.

Fig. R4 Variation in normalized mode order, percentage sum of first 100 modes, and percentage sum of first 1000 modes with displacement distance under a wide-range low-resolution measurement (0–10 mm).

Fig. R5 Percentage distribution of LP mode groups (first 1000 modes grouped in sets of 100) versus displacement distance under the 0-30mm wide-range low-resolution measurement, along with the light field distribution coupled into the fiber.

The corresponding simulations for Fig. 2f and Fig. 2g under wafer microscopic images

have been updated. Due to the minimal changes in mode group proportions under small displacements, Fig. 2f still adopts a large displacement interval (ranging from 0 mm to 30 mm) with simulation analysis performed every 10 mm, as shown in Fig. R5 below. Fig. 2g presents the intensity proportion of the top 20 modes at a detection displacement of 0 nm, as illustrated in Fig. R6. From the trend observed in Fig. 2f, we can similarly conclude that as the displacement distance increases, the energy of the coupled optical field shifts toward lower-order modes.

Fig. R6 Proportion of intensity for the first 20 modes and spatial distribution of modes with high intensity proportion under 0 nm displacement detection.

Overall, we wish to emphasize that through modal decomposition of the optical field under different displacements, a certain inherent relationship exists between the displacement of the detection target and the mode composition within the multimode fiber, ultimately influencing the distribution of the output speckle field. Although some analyzed variables may exhibit fluctuations or even opposite trends under small displacements, this intrinsic relationship remains present. This part of the study provides a physical basis and a degree of interpretability for the displacement recognition neural network discussed later in the manuscript.

Following your suggestion, we have replaced Fig. 2k in the main text with updated results showing the normalized mode order and the variation in mode component proportions under small displacements and high resolution. This effectively supports the experimental results presented subsequently, enhancing the overall coherence and self-consistency of the article. The specific modifications are as follows:

(From page 4, line 147)

Fig. 2a shows the physical simulation model. Fig. 2b presents the input optical field chosen for
 the simulation. Since the output speckle field is also influenced by the two-dimensional spatial
 structure of the detected target, we selected a wafer micrograph, which contains both high-frequency
 and low-frequency structures, to excite higher-order modes and obtain universal results. Fig. 2c and
 2d show the fiber input facet and field at the core area after free space transmission and fiber-end-
 ball transformation, respectively. We performed mode decomposition on the input optical field and
 selected the first 1000 transverse mode components for reconstruction, as shown in Fig. 2e. It can
 be observed that Fig. 2e contains most of the structures from Fig. 2d. We further analyze the energy
 percentage of the mode components in the input optical field and their relationship with the
 displacement. The first 1000 modes are divided into 10 groups in sequential order, with each group
 containing 100 modes. The intensity percentage of each group is the sum of the intensity percentages of
 the 100 modes in the group. As the detected displacement increases from 0 mm to 30 mm, it can be
 found from Fig. 2f that the share of the fundamental mode component gradually increases, and more

(From page 5, line 165)

Fig. 2 Physical model of MMF and simulation results at different displacements. a, Different combinations of transmission modes of MMFs are coupled at different detected displacement z , forming different speckles. b-e, The detected target, fiber input facet, field at the core area, and the recombination result of the first 1000 modes. f, The first 1000 modes are divided into 10 groups in sequential order, with each group containing 100 modes. The intensity percentage of each group is the sum of the intensity percentages of the 100 modes in the group. The intensity percentage of these ten groups and the combination field at 0 mm, 10 mm, 20 mm, and 30 mm displacements. g, Percentage of the first 20 modes and the distribution of the main modes at the detected displacement of 0 nm. h, Normalized average mode order (blue left axis), percentage of the first 100 modes (orange right axis) and the first 1000 modes (red right axis) at different detected displacement (0-990nm, 100 sampling points).

(From page 6, line 183)

average mode order of the coupled mode combination. This coefficient reflects to some extent the
relationship between the displacement and the output speckle distribution. The result in Fig. 2h
shows that as the displacement distance increases, the normalized average mode order shows a
monotonically increasing trend, while the proportions of both the first 100 modes and the first 1000
LP modes demonstrate monotonically decreasing trends. The law observed here is opposite to that
in Fig. 2f. It may be attributed to the detection target not reaching the equivalent focal plane of the
fiber-end-ball. Partial defocusing effects will cause a reduction of high-frequency components⁴³,
resulting in a higher proportion of fundamental modes. As the displacement distance increases, the
target approaches the equivalent focal plane of the fiber-end-ball. The energy of the coupled optical
field shifts toward higher-order modes. To further investigate the patterns under large displacement
ranges, we have supplemented the simulation results of wafer microscopic images within a large
displacement measurement range of 0–10 mm in the Supplementary Information. In this case, the
results are consistent with the laws of Fig. 2f. More details of the physical model and results are
presented in Supplementary Note 1, Supplementary Table 1, and Supplementary Fig.S1, S2 and S3.

As well, we have added the variation curve in normalized mode order, percentage sum of first 100 modes, and percentage sum of first 1000 modes with displacement distance under larger displacement ranges to the Supplementary Information, making the manuscript more comprehensive. The specific modifications are as follows:

(Supplementary Information, from page 4, line 119)

Fig. S1 shows simulation architecture for optical field evolution of multimode fiber probes.
 Simulation results of the physical model at different displacements ($z=0\text{mm}$, $z=10\text{mm}$, $z=20\text{mm}$,
 $z=30\text{mm}$) are shown in Fig. S2. We select Lenna image here, which also contains both high-
 frequency and low-frequency structures.[Ⓢ]
 Ⓢ
 In order to characterize the overall mode distribution and the higher-order mode share of different
 displacement, the normalized average mode order is defined:[Ⓢ]
 Ⓢ
$$O_n = \frac{\sum_i P_i(i) \times i}{\sum_i P_i(i)} \quad (S13)$$
[Ⓢ]
 Ⓢ
 To further investigate the patterns under large displacement ranges, we used wafer microscopic
 images to simulate within a large displacement measurement range of 0–10 mm. Sampling points
 were taken at 1 mm intervals, and scatter plots of the normalized mode order, the percentage sum
 of first 100 LP modes, and percentage sum of first 1000 LP modes were plotted along with fitted
 curves, as shown in Fig. S3. It can be observed that, although fluctuations exist in the normalized
 mode order and the percentage sum of first 100 and first 1000 LP modes at certain positions, an
 overall monotonic trend is evident. Within the large displacement range, the proportions of both
 the first 100 and the first 1000 LP modes exhibit an increasing trend, while the normalized average
 mode order shows a decreasing trend. These results are consistent with Fig. 2f. The normalized
 modes coefficient are also shown in Table S1.[Ⓢ]
 Ⓢ
 In order to show a clear trend of change, we increase the interval of displacement to 5mm and
 simulate the normalized average mode order at different displacements from 0 to 40mm. The result
 shows that when the displacement increases, the normalized modes coefficient indicates an overall
 decreasing trend as shown in Table S2.[Ⓢ]
 Ⓢ
 Overall, we wish to emphasize that through modal decomposition of the optical field under
 different displacements, a certain inherent relationship exists between the displacement of the
 detection target and the mode composition within the multimode fiber, ultimately influencing the
 distribution of the output speckle field. Although some analyzed variables may exhibit fluctuations
 or even opposite trends under small displacements, this intrinsic relationship remains present. This
 part of the study provides a physical basis and a degree of interpretability for the displacement
 recognition neural network discussed later in the manuscript.[Ⓢ]

(Supplementary Information, from page 25, line 933)

(Supplementary Information, from page 52, line 1118)

1117	Supplementary Table																								
1118	Table S1. Results of the simulated normalized modes coefficient indicate (Wafer)																								
	Distance/mm012345678910O_n11.9011.5311.8811.309.4310.429.718.949.339.4310.04	Distance/mm	0	1	2	3	4	5	6	7	8	9	10	O_n	11.90	11.53	11.88	11.30	9.43	10.42	9.71	8.94	9.33	9.43	10.04
Distance/mm	0	1	2	3	4	5	6	7	8	9	10														
O_n	11.90	11.53	11.88	11.30	9.43	10.42	9.71	8.94	9.33	9.43	10.04														
1119	Table S2. Results of the simulated normalized modes coefficient indicate (Lenna)																								
	Distance/mm0510152025303540O_n24.3322.3218.2217.3318.3115.8311.459.9910.14	Distance/mm	0	5	10	15	20	25	30	35	40	O_n	24.33	22.32	18.22	17.33	18.31	15.83	11.45	9.99	10.14				
Distance/mm	0	5	10	15	20	25	30	35	40																
O_n	24.33	22.32	18.22	17.33	18.31	15.83	11.45	9.99	10.14																
1120																									

Comment 5: The authors demonstrated the robustness of the displacement recognition network under various fiber bending conditions. But the authors did not evaluate whether the inverse transmission matrix imaging network can maintain image reconstruction quality under various fiber bending conditions.

Response:

Thank you for your suggestions regarding the bending robustness of the system. In the manuscript, we demonstrated that, even under varying bending radii of the optical fiber, high-precision and high-accuracy displacement recognition can still be achieved using joint learning.

Due to space limitations, we did not include an evaluation of whether the optimized the inverse transmission matrix imaging network could maintain imaging performance under various bending conditions. Nevertheless, such feasibility exists, and we take this opportunity to provide a detailed explanation of our supplementary experiments and results.

imaging systems constructed based on transmission matrices are often criticized for their poor robustness. This is particularly true for fiber-optic imaging systems, where achieving consistent imaging performance under various bending conditions remains a significant challenge in the research community. In our previous work, we partially addressed this issue by employing joint learning to enable imaging of simple letters and patterns under different bending radii (Ref: Laser & Photonics Reviews, 16, 2100724, 2022). Specifically, the fiber was bent at a certain distance from the fiber-end sphere, and the bending radius was recorded. It should be noted that, to avoid introducing stress into the multimode fiber, adjusting the bending radius of the front segment simultaneously affected the morphology of the rear segment.

In the experiment, the bending diameter of the fiber was gradually adjusted from 14 cm to 22 cm in increments of 2 cm. At each interval, 8,000 sets of training data and 500 sets of validation data were collected. The bending radius was not further reduced to prevent excessive higher-order mode loss and a decrease in the number of guided modes, both of which could compromise imaging quality. Joint training and testing were performed on five datasets corresponding to their respective bending radii. The experimental concept and test results are illustrated in Fig. R7.

[editorial note: third party material redacted]

Fig. R7 Joint learning results of the integrated multimode fiber imaging system under different bending diameters. a-e, Joint learning reconstruction results under bending states ranging from 14 to

22 cm, respectively.

At most sampling points within the tested bending range, the structural similarity index (SSIM) of the reconstructed images remained above 0.3. This indicates that, although imaging quality degraded compared to systems with a fixed bending radius, the joint learning approach enabled the fiber probe to maintain imaging capability even within a certain range of morphological variations caused by bending. The system successfully reconstructed handwritten digits and geometric patterns, and partially recovered contour information of complex grayscale images.

The aforementioned results only present the joint learning outcomes at five bending radii. As the number of sampled radii increases, imaging under flexible bending within a certain range could theoretically be achieved. However, this would likely require larger-scale computational resources and higher costs due to the increased training data.

On the other hand, while the joint learning method has improved the bending resistance of the all-fiber system to some extent, the imaging quality has degraded compared to that under fixed bending conditions. To enhance bending robustness without compromising imaging quality, one feasible solution is the construction of a "bending-state–transmission-matrix library," which will be introduced below.

To simultaneously improve the bending robustness and maintain high-fidelity image reconstruction in an integrated multimode fiber imaging system, an additional network can be introduced to construct a "Bending-state–transmission-matrix library." In fact, a similar exhaustive pre-training approach has been applied in "light-field encoding" imaging (Ref: *Nature Photonics*, 17, 679-687, 2023), enabling endoscopic examination in complex intracavitary environments—such as those with long-term presence of fluids, extended cavities, and temperature variations—in tissues like sheep small intestine and pig esophagus.

Fig. R8 Confusion matrix for bending state recognition and VGG bending state recognition network architecture. a, Confusion matrix for bending state recognition. **b**, The architecture of VGG bending state recognition network.

By employing two separate networks for bending state recognition and inverse transmission matrix optimization for imaging tasks, high-quality image reconstruction under different bending radii is achieved. Specifically, the five datasets corresponding to bending diameters from 14 cm to 22 cm were labeled with their respective bending radii and fed into a VGG classification network. The captured speckle data were randomly shuffled, with 70% used for training, 10% for validation, and 20% for testing. As shown in Fig. R8a, the bending radius identification results are presented, with the vertical axis representing the actual bending radius

and the horizontal axis indicating the predicted bending radius. It can be observed that all data points align along the diagonal, indicating accurate identification.

Fig. R8b illustrates the architecture of the VGG classification network used for bending radius recognition. The input to the VGG bending state identification network is a diffuse speckle pattern compressed to a resolution of 128×128 . The network first processes the input through a 3×3 convolutional layer, followed by a hyperbolic tangent (tanh) activation layer and a 2×2 max pooling layer. This structure is repeated sequentially, increasing the channel depth progressively from 32 to 256, while maintaining consistent kernel size, activation, and pooling operations throughout. The output is then flattened and fed into a fully connected layer with a size of 1×16384 , using tanh as the activation function. Two subsequent fully connected layers gradually reduce the number of elements from 16384 to N , which corresponds to the number of bending sample categories. The model was trained for 50 epochs using the Adam optimizer with an initial learning rate of 1×10^{-3} , until convergence was achieved.

On the other hand, an inverse transmission matrix (ITM) optimization network was utilized to reconstruct images from data collected under different bending radii and to further extract the optimized inverse transmission matrices. These ITMs were labeled with their corresponding bending radii and integrated to construct a "Bending-state transmission matrix library." The specific testing workflow is illustrated in Fig. R9b. When an output speckle pattern captured under an unknown bending radius is acquired, it is first input into the VGG bending-state recognition network to identify the current bending state of the system. Based on the output of the VGG network, the corresponding inverse transmission matrix associated with the identified bending state is retrieved from the bending-state transmission matrix library and used for image reconstruction. The reconstruction results are shown in Fig. R9c.

The above results demonstrate the feasibility of the method using five sampling instances with bending applied at a single location and in one direction. However, it is expected that by increasing the number of sampling points, reducing the interval between bending diameters, and performing repeated sampling in multiple directions, this approach could theoretically enable image reconstruction under flexible bending of a single multimode fiber probe. Of course, handling such a large volume of data would require higher-speed acquisition efficiency and more powerful parallel computing capabilities.

[editorial note: third party material redacted]

Fig. R9 Bend-resistant imaging scheme based on bending state recognition networks and transmission matrix libraries. **a**, Computing the ITM for a corresponding bending state by acquiring multiple speckle images under different bending conditions. **b**, Implementing a high-quality bend-resistant imaging workflow using a bending state recognition network and invoking the "Bending state-transmission matrix library." **c**, Image reconstruction results.

In response to your suggestion, we will add an introduction to the "Bending State-Transfer Matrix Library" method and imaging results under different bending states to the Supplementary Information. The specific modifications are as follows:

(Supplementary Information, from page 12, line 459)

To validate the system's imaging capabilities under different bending states, we conducted
verification experiments. To simultaneously improve the bending robustness and maintain high-
fidelity image reconstruction in an integrated multimode fiber imaging system, an additional
network can be introduced to construct a "Bending-state-transmission-matrix library." In fact, a
similar exhaustive pre-training approach has been applied in "light-field encoding" imaging (Ref.
27), enabling endoscopic examination in complex intracavitary environments—such as those with
long-term presence of fluids, extended cavities, and temperature variations—in tissues like sheep
small intestine and pig esophagus.⁴²

(Supplementary Information, from page 13, line 468)

By employing two separate networks for bending state recognition and inverse transmission matrix
optimization for imaging tasks, high-quality image reconstruction under different bending radii is
achieved. Specifically, the five datasets corresponding to bending diameters from 14 cm to 22 cm
were labeled with their respective bending radii and fed into a VGG classification network. The
captured speckle data were randomly shuffled, with 70% used for training, 10% for validation, and
20% for testing. As shown in Fig. S16a, the bending radius identification results are presented,
with the vertical axis representing the actual bending radius and the horizontal axis indicating the
predicted bending radius. It can be observed that all data points align along the diagonal, indicating
accurate identification.⁴²

⁴²

Fig. S16b illustrates the architecture of the VGG classification network used for bending radius
recognition. The input to the VGG bending state identification network is a diffuse speckle pattern
compressed to a resolution of 128×128. The network first processes the input through a 3×3
convolutional layer, followed by a hyperbolic tangent (tanh) activation layer and a 2×2 max-
pooling layer. This structure is repeated sequentially, increasing the channel depth progressively
from 32 to 256, while maintaining consistent kernel size, activation, and pooling operations
throughout. The output is then flattened and fed into a fully connected layer with a size of 1×16384,
using tanh as the activation function. Two subsequent fully connected layers gradually reduce the
number of elements from 16384 to N, which corresponds to the number of bending sample
categories. The model was trained for 50 epochs using the Adam optimizer with an initial learning
rate of 1×10^{-3} , until convergence was achieved.⁴²

⁴²

On the other hand, an inverse transmission matrix (ITM) optimization network was utilized to
reconstruct images from data collected under different bending radii and to further extract the
optimized inverse transmission matrices. These ITMs were labeled with their corresponding
bending radii and integrated to construct a "Bending-state transmission matrix library." The
specific testing workflow is illustrated in Fig. S16d. When an output speckle pattern captured
under an unknown bending radius is acquired, it is first input into the VGG bending-state
recognition network to identify the current bending state of the system. Based on the output of the
VGG network, the corresponding inverse transmission matrix associated with the identified
bending state is retrieved from the bending-state transmission matrix library and used for image
reconstruction. The reconstruction results are shown in Fig. S16e.⁴²

⁴²

The above results demonstrate the feasibility of the method using five sampling instances with
bending applied at a single location and in one direction. However, it is expected that by increasing
the number of sampling points, reducing the interval between bending diameters, and performing
repeated sampling in multiple directions, this approach could theoretically enable image
reconstruction under flexible bending of a single multimode fiber probe. Of course, handling such
a large volume of data would require higher-speed acquisition efficiency and more powerful
parallel computing capabilities.⁴²

(Supplementary Information, from page 38, line 1016)

[editorial note: third party material redacted]

Comment 6: There is no discussion for Fig. 2g. And why there are different fluctuation situations in the excited modal orders of Fig. 2f and 2g.

Response:

Thank you for pointing out this oversight. We indeed did not provide an explanation for Fig. 2g in the main text and only described it briefly in the caption. However, we would like to clarify that the differences in mode composition between Fig. 2f and Fig. 2g arise from the fact that they represent different scenarios and data integration methods. We acknowledge that this may have caused confusion, and we will now provide a detailed explanation.

First, we elaborate on the meaning of Fig. 2g. This figure shows the energy proportions of the top 20 modes after modal decomposition when the detection target is at a displacement of 0 nm. We have also annotated the spatial distribution patterns corresponding to certain modes with higher energy proportions. The purpose of this figure is to illustrate the distribution of lower-order modes that dominate in energy contribution. Notably, it can be observed that modes with circularly symmetric distributions, such as LP_{01} , LP_{02} , and LP_{03} , generally exhibit higher energy proportions.

To avoid potential confusion among readers, we have added a detailed description and interpretation of Fig. 2g in the corresponding section of the manuscript, as follows:

(From page 4, line 162)

order mode components in the speckle field, corresponding to a decrease in the intensity of the
speckle periphery. Fig. 2g shows the percentage of the first 20 modes and the distribution of the
main modes at the detected displacement of 0 nm. It can be observed that LP patterns exhibiting
rotational symmetry (LP_{0n}) generally account for a higher proportion of intensity.

Furthermore, we would like to clarify the differences between Fig. 2g and Fig. 2f. As mentioned above, Fig. 2g displays the energy distribution of low-order modes when the target is positioned at 0 nm displacement. Each bar in this figure represents an individual LP mode.

In contrast, Fig. 2f illustrates changes in modal energy composition across different displacement positions for the same target image. It should be emphasized that the vertical axis in Fig. 2f represents the combined energy proportion of multiple modes. For example, Bar 1 corresponds to the sum of energy proportions of modes 1 to 100, while Bar 10 represents the sum of modes 901 to 1000. Each bar in Fig. 2f thus represents a group of 100 modes, unlike Fig. 2g where each bar corresponds to a single mode.

The use of mode groups in Fig. 2f aims to present a more statistically averaged result. Variations in energy proportions across different displacements reflect the overall trend of coupling efficiency between low-order and high-order modes.

To prevent any potential confusion among readers, we have revised the manuscript to provide a clearer emphasis on the data processing methodology, as follows:

(From page 4, line 155)

percentage of the mode components in the input optical field and their relationship with the
displacement. The first 1000 modes are divided into 10 groups in sequential order, with each group
containing 100 modes. The intensity percentage of each group is the sum of the intensity percentages of
the 100 modes in the group. As the detected displacement increases from 0 mm to 30 mm, it can be
found from Fig. 2f that the share of the fundamental mode component gradually increases, and more

Comment 7: A detailed description of the inverse transmission matrix optimization network should be provided. Or relevant reference to it should be provided, if any.

Response:

Thank you for your suggestion. We acknowledge that the inverse transmission matrix was not explained in detail in the manuscript. On one hand, the primary focus of this work is to achieve high-precision displacement measurement using the super-oscillatory speckle patterns generated by multimode fibers. The imaging function of the system is primarily intended to demonstrate its potential applications in fields such as minimally invasive medicine. On the other hand, we assumed that the inverse transmission matrix is already widely recognized and commonly used in the field of multimode fiber imaging. Nevertheless, we agree that adding a description of the inverse transmission matrix in the imaging section is necessary to help readers better understand the underlying principle of the network. Therefore, we will include a relevant introduction in the manuscript. A brief overview of the inverse transmission matrix method is provided below:

The Transmission Matrix (TM) method is primarily used to study wave behavior as it propagates through different media or structures. It decomposes a complex system into a series of simple layers or units, with each layer defined by a transmission matrix that describes its effect on the incident wave. The overall transmission characteristics of the system can then be obtained by multiplying these matrices. It serves as a tool for modeling light propagation through strongly scattering media and multimode fibers, and is also one of the methods for computational image reconstruction. Since it accounts for processes such as reflection, scattering, and interference of light in scattering media, it can accurately describe how scattering media transform optical fields.

Fig. R10 illustrates the concept of input and output optical fields in a multimode fiber. In Hilbert space, the input optical field and the output speckle can be idealized as an $m \times m$ complex matrix and an $n \times n$ complex matrix, respectively. In the Cartesian coordinate system, each pixel of the input image can be treated as an input mode (with other pixel values set to zero), resulting in $N_i = m^2$ input modes and $N_o = n^2$ output modes. Their field distributions are denoted as \mathbf{E}_{in} and \mathbf{E}_{out} , respectively.

[editorial note: third party material redacted]

Fig. R10 Correspondence between input and output light fields of multimode fiber in the transmission matrix description framework.

The specific quantitative relationship between the output speckle field \mathbf{E}_{out} and the input optical field \mathbf{E}_{in} is generally unknown. However, it can be formally represented using a Green's function approach:

$$E_{out}(x, y) = \frac{i}{\lambda} \iint_{S_{in}} E_{in}(\xi, \eta) g(\xi, \eta, x, y) d\xi d\eta \quad (1 - 2)$$

where $g(\xi, \eta, x, y)$ is the Green's function. This expression can be discretized by decomposing it into contributions from individual input modes:

$$E_{out}^n(x, y) = \frac{i}{\lambda} \sum_{n=1}^N E_{in}^m \iint_{S_{in}^n} g(\xi, \eta, x, y) d\xi d\eta \quad (1-3)$$

where E_{in}^m and E_{out}^n are the complex amplitudes of the m-th input mode and the n-th output mode, respectively. The field transformation coefficient t_{mn} is defined as:

$$t^{mn} = \frac{i}{\lambda} \iint_{S_{in}^n} g(\xi, \eta, x, y) d\xi d\eta \quad (1-4)$$

Thus, the output field can be expressed as:

$$E_{out}^m(x, y) = \sum_{n=1}^N E_{in}^n t^{mn} \quad (1-5)$$

Converting the input and output images into one-dimensional vectors, where each element represents a complex amplitude, allows the above equation to be expressed in matrix form:

$$\begin{bmatrix} e_{out}^1 \\ e_{out}^2 \\ \vdots \\ e_{out}^{n^2} \end{bmatrix} = \begin{bmatrix} t_{11} & t_{12} & \cdots & t_{1m^2} \\ t_{21} & t_{22} & \cdots & t_{2m^2} \\ \vdots & \vdots & \ddots & \vdots \\ t_{n^2 1} & t_{n^2 2} & \cdots & t_{n^2 m^2} \end{bmatrix} \begin{bmatrix} e_{in}^1 \\ e_{in}^2 \\ \vdots \\ e_{in}^{m^2} \end{bmatrix} \quad (1-6)$$

where $E_{in} = [e_{in}^1 \ e_{in}^2 \ \cdots \ e_{in}^{m^2}]^T$ is the input light field, $E_{out} = [e_{out}^1 \ e_{out}^2 \ \cdots \ e_{out}^{n^2}]^T$

is the output light field, and e_{in}^n and e_{out}^m are the complex amplitudes of each pixel in the input image and output speckle, respectively.

Therefore, if the TM of the MMF can be determined, the relationship between the input image field and the output speckle field can be described as:

$$E_{out} = T \cdot E_{in} \quad (1-7)$$

All possible input and output vectors span N_i dimensional and N_o dimensional Hilbert spaces, respectively, and the transmission matrix establishes a linear mapping between these two spaces.

However, conventional TM-based fiber imaging systems not only suffer from poor robustness but also often require access to both ends of the fiber to perform point-scanning imaging. To address the limitation of requiring dual-end access for TM measurement and to improve the data adaptability of neural network-based imaging, Daniele Faccio et al. from the University of Glasgow combined the TM method with a neural network architecture in 2019 (Ref: Nature Communications, 10, 2029, 2019). They constrained the network using the input-output matrix mapping relationship. Specifically, they used a two-network combination to reconstruct the input image: the first network retrieves the input pattern from the output intensity distribution, and the second network maps the inferred input pattern back to the original speckle distribution provided to the first network. Essentially, the first network learns the backward mapping from the output intensity to the input pattern, while the second network

learns the forward mapping. This approach enabled the reconstruction of natural grayscale scenes from the ImageNet dataset with strong generalization capabilities.

For the highly integrated in-situ fiber sensing system presented in this work, conventional TM measurement methods generally require complex optical setups and reference beams, which are difficult to implement in our all-fiber system. Therefore, inspired by the image reconstruction architecture of Daniele Faccio et al., we combine the TM method with a neural network architecture to directly optimize and solve for the inverse of the transmission matrix. This is because the inverse matrix is more critical for image reconstruction than the TM itself, and this approach effectively avoids the need for direct inversion of large matrices. Whether in the frequency domain or the spatial domain, solving the TM requires acquiring a set of diverse input responses. Specifically, the optical path between the input image and the captured speckle field can be regarded as an integrated transmission system. Thus, the relationship between the input image field of the MMF imaging system and the intensity distribution captured by the camera can be expressed as:

$$I_i = |T^{-1}E_s|^2 \quad (1 - 8)$$

Here, I_i and E_s represent the intensity distribution of the detected image and the amplitude distribution of the output speckle pattern (measured as the square root of the speckle intensity pattern), respectively. T is the transmission matrix of the all-fiber imaging system.

After capturing a large number of output speckles with the camera, their square roots must first be extracted. This converts the intensity data into amplitude data. Since the transmission matrix is complex, the imaginary part must be set to zero.

$$\mathbf{X} = \text{sqrt}(\mathbf{I}) + 0 \cdot j = \begin{bmatrix} \sqrt{i_{1,1}} + 0 \cdot j & \cdots & \sqrt{i_{1,m}} + 0 \cdot j \\ \vdots & \ddots & \vdots \\ \sqrt{i_{m,1}} + 0 \cdot j & \cdots & \sqrt{i_{m,m}} + 0 \cdot j \end{bmatrix} \quad (1 - 9)$$

Here, \mathbf{I} represents the speckle intensity distribution captured by the camera. After supplementing the imaginary part, \mathbf{X} becomes the constructed speckle complex amplitude distribution. $i_{n,m}$ is the element at row n and column m in the complex amplitude distribution. It is assumed that the speckle field input to the network is a square matrix with m rows and m columns. This is then converted into a column vector.

Next, the Inverse Transmission Matrix (ITM) is initialized using the ‘‘Glorot Uniform’’ method. It is then multiplied by the speckle matrix as follows:

$$\mathbf{Y}_R = |\mathbf{T}^{-1}\mathbf{X}|^2 \quad (1 - 10)$$

The square modulus of the product matrix yields the vector for reconstructing the image, namely \mathbf{Y}_R .

Express the following equation in matrix form:

$$\begin{bmatrix} \mathcal{Y}_{R(1,1)} \\ \vdots \\ \mathcal{Y}_{R(n,n)} \end{bmatrix} = \left\| \begin{bmatrix} a_{1,1} + b_{1,1}j & \cdots & a_{1,m^2} + b_{1,m^2}j \\ \vdots & \ddots & \vdots \\ a_{n^2,1} + b_{n^2,1}j & \cdots & a_{n^2,m^2} + b_{n^2,m^2}j \end{bmatrix} \begin{bmatrix} \sqrt{i_{1,1}} + 0 \cdot j \\ \vdots \\ \sqrt{i_{m,m}} + 0 \cdot j \end{bmatrix} \right\|^2 \quad (1 - 11)$$

[editorial note: third party material redacted]

Fig. R11 Image reconstruction algorithm based on inverse transmission matrix optimization. a, Mathematical correspondence between the inverse transmission matrix, acquired speckle field, and reconstructed image. **b,** Structure of the inverse transmission matrix optimization network.

In the training architecture of neural networks, after each matrix multiplication operation is completed, the cosine similarity between the reconstructed image \mathbf{Y}_R and the true image \mathbf{Y} (from the training set) is calculated.

$$L = \frac{\sum_{i=1}^{n^2} (y_{Ri} \times y_i)}{\sqrt{\sum_{i=1}^{n^2} (y_{Ri})^2} \times \sqrt{\sum_{i=1}^{n^2} (y_i)^2}} \quad (1 - 12)$$

The cosine similarity is used as the loss function to optimize the elements of the inverse transmission matrix via gradient descent until an approximate solution within the preset target range is obtained.

For the inverse transmission matrix optimization network, the detailed architecture is shown in Fig. R11b. First, the speckle field is decomposed into its complex representation. It is then fed into a complex-valued dense layer. After initialization and normalization, the normalized complex inverse transmission matrix is multiplied by the input speckle field. The output is compared with the ground truth, and the neural network feedback framework is used to further optimize the inverse transmission matrix. The final output is a reconstructed image that meets the preset performance metrics.

The ITM method used here, which relies on dataset training, represents an approach between traditional full TM solving and deep learning neural networks that make no assumptions about the system. Compared to solving the full TM, it eliminates the need for a reference beam and phase information collection. Compared to traditional deep learning methods, it incorporates physical constraints by restricting the system's representation to the form of a TM under this specific physical scenario, thereby giving the network a concrete form. This achieves simple, efficient, and high-quality image transmission.

To help readers better understand the principle of the inverse transmission matrix and the system's imaging method, we have added the following supplementary description in the corresponding section of the manuscript:

(From page 12, line 379)

**Image reconstruction with fiber probe**⁴
In recent years, multimode fibers have attracted considerable interest in the field of ultra-thin
flexible endoscopy. With their larger information capacity and smaller size, they are expected to
replace fiber bundles and provide powerful tools for minimally-invasive surgery⁴⁹ and deep brain
neuroimaging^{25,26,28,29}. As mentioned earlier, the speckles at the fiber output also contain information
about the input optical field or the target structure. This inverse problem can be solved by either
determining the system's transmission matrix or building neural networks. Here, image
reconstruction is achieved using an ITM network. This approach combines the transmission matrix
method with a neural network architecture⁵⁰. Compared to solving the full TM, it eliminates the
need for a reference optical path and does not require phase information collection. In contrast to
traditional deep learning methods, it constrains the neural network through the input-output matrix
mapping relationship, resulting in stronger generalization capability. Based on this, we
experimentally demonstrated that our multimode fiber detection system can simultaneously perform
both positioning and imaging, as shown in Fig. 6a.⁴

(From page 17, line 559)

**Neural network architecture**

The diffuse speckle field, compressed into a 128×128 pixels resolution, is initially processed by the
VGG displacement recognition network. The input first traverses a 3×3 convolutional layer,
followed by a hyperbolic tangent (tanh) activation function and a 2×2 max-pooling layer. This
sequence is repeated iteratively, with the channel depth incrementally doubled from 32 up to 256.
Throughout these iterations, the kernel size, activation function, and pooling parameters remain
constant. The resulting output is subsequently flattened and input into a fully connected (FC) layer
of dimensions 1×16384 , where the tanh activation function is applied. Following this, two additional
FC layers sequentially reduce the dimensionality from 16384 to the number of displacement samples
(N).

For the inverse transmission matrix optimization network, the compressed speckle is first
complexified, as the transmission matrix is a complex matrix. It is then fed into the complex dense
layer. Here, an initialized and normalized complex inverse transmission matrix is multiplied with
the input diffuse speckle. Since the partial speckle distribution also contains all the information of
the input image, the reconstructed image can be obtained with a sufficient amount of information.
The output is then compared with the ground truth and the inverse transfer matrix is further
optimized using a neural network feedback framework. The final output are reconstructed images
after convergence of the loss function.

Comment 8: Whether the sequential order of the dual-network in Fig. 6b is correct? As discussed by the authors, whether the neural network should sense the displacement first and then find the corresponding ITM in the displacement dependent ITM library.

Response:

Thank you for bringing this to our attention. We would like to clarify that Fig. 6 was not intended to illustrate the computational sequence of the network, but rather to demonstrate the data processing flow and the capability of the dual-network architecture to simultaneously achieve both displacement localization and imaging functions. Your careful reading and kind reminder made us aware that the original presentation could potentially cause misunderstanding or confusion among readers.

You are absolutely correct: when both localization and imaging functions are required, we must first construct a displacement-dependent ITM library. During testing, this library is used after displacement recognition, after which the corresponding ITM is retrieved to carry out image reconstruction.

Following your suggestion, we have swapped the order of the example patterns for the ITM and the displacement recognition neural network, as shown below.

(From page 13, line 400)

[editorial note: third party material redacted]

However, we wish to note that this figure only briefly illustrates the data workflow and the two functions realized by the dual-network system. For more detailed implementation strategies and computational sequences, we refer the readers to Movie 2 and the Supplementary Information.

Response to Reviewer 2

This manuscript presents an in situ displacement measurement technology based on a multimode fiber probe generating superoscillatory speckles. Combined with a VGG convolutional neural network, the system achieves non-contact detection at 10 nm resolution and 99.95% accuracy, demonstrating robust performance under varying fiber bending radii and materials via joint learning. The lens-free imaging capability of the system is also experimentally validated. However, these results are insufficient to guarantee publication in Nature Communications for several reasons. (1) First, multimode fiber probes have been vastly studied for endoscopic imaging, and extending their application to distance measurement here involves key technologies already widely discussed in endoscopy, lacking sufficient technical originality. (2) Second, for displacement measurement (especially longitudinal displacement), white-light interferometry-based techniques already achieve sub-nanometer precision, making the reported 10 nm accuracy in this work technically unadvanced. (3) Additionally, the scheme requires object-specific training, limiting its generality as a universal device.

Thank you for your thorough review of our work and for raising valuable questions and concerns. In response to the main points you have questioned, we have conducted extensive investigation and comparative analysis, and we provide the following clarification.

(1) First, as you rightly pointed out, multimode fibers (MMFs) have been widely studied in recent years due to their extremely high information capacity and flexible, compact physical characteristics, and they are considered to have broad application prospects in the field of ultra-thin endoscopic imaging. However, most research has focused solely on imaging functions and generally requires complex coherent optical setups. We contend that the applications of multimode fibers in the field of imaging and in displacement measurement (the primary focus of this work) rely on fundamentally distinct technical approaches. The former addresses tasks related to image reconstruction and regression, whereas the latter is oriented toward modal identification and classification. The underlying physical mechanisms also differ significantly: imaging primarily involves optical field evolution, while displacement metrology leverages the super-oscillatory characteristics of speckle patterns. In the field of displacement metrology, MMFs are commonly studied for transverse deformation and pressure sensing. To the best of our knowledge, however, there is no existing research on MMF systems for non-contact displacement metrology of samples themselves, particularly in the domain of longitudinal displacement measurement. Their potential in phase detection or displacement sensing remains largely untapped. We believe this study pioneers a novel pathway for employing multimode fibers in the domain of high-precision metrology beyond conventional imaging, demonstrating exciting potential and broad prospects for a variety of metrological applications.

(2) Secondly, traditional displacement metrology methods all have significant limitations. For example, Michelson/Mach-Zehnder interferometers (Fig. R12a) are highly sensitive to airflow disturbances, platform vibrations, and electromagnetic noise. Traditional optical interferometers are bulky and require optical path steering components in the monitoring environment, making flexible detection challenging. Probe sizes of eddy current sensors (Fig. R12b) and capacitive sensors (Fig. R12c) are still limited to tens of millimeters, which restricts

their use to conductive surfaces. Emerging techniques such as orbital angular momentum (Fig. R12d) and single-cavity loss spectroscopy offer high resolution and signal-to-noise ratio, but they require access to both sides of the target, which is often impractical. In contrast, the in-situ non-contact measurement approach using MMFs proposed in our study not only enables single-ended non-contact access deep within structures (Fig. R12e) but also features a simple optical setup (Fig. R13a) and low cost. Furthermore, a major advantage of our method over traditional MMF sensing and other displacement metrology techniques is the use of deep learning to achieve high-precision and high-accuracy displacement recognition across various sample patterns.

[editorial note: third party material redacted]

Fig. R12 Probe of different high-precision displacement measurement techniques. a, Traditional interferometer with fiber optic probe. (Ref: *Sensors and Actuators A: Physical*, 190, 106-126, 2013) **b**, Probe of eddy current sensors. (Ref: *IEEE Sens. J.*, 7, 1538-1545, 2007) **c**, Capacitive probe. (Ref: *IEEE Transactions on Instrumentation and Measurement*, 60, 2730-2737, 2011) **d**, Setup of orbital angular momentum. (Ref: *Nat. Commun.*, 15, 8391, 2024) **e**, micro-probe of this work.

Regarding the white-light interferometry technique you mentioned, we acknowledge that it is indeed a mature solution in the industry with high displacement resolution. High-end products can even achieve sub-nanometric resolution (Table R1). However, it still has significant limitations and scenarios where it cannot be effectively applied. The optical structure and principle of a traditional white-light interferometer are shown in the Fig.R13b below. It requires a broadband and stable white-light source, a color camera, and a piezoelectric ceramic controller. Furthermore, the optical path must incorporate multiple optical components and beam splitters to form interference patterns. The output probe light must illuminate the sample at a fixed angle, which requires the target to be exposed to the environment. For samples located inside equipment or within narrow pipelines, the light beam cannot be flexibly bent to achieve detection, limiting the applicable scenarios of such devices. Additionally, similar to traditional interferometric ranging systems, the coherent optical path is highly sensitive to airflow disturbances, platform vibration, and electromagnetic noise. In contrast, the system proposed in our manuscript employs neither a coherent optical path nor complex optical components, resulting in extremely low cost. It is capable of performing in-situ non-contact measurements deep within narrow or internal systems. In terms of detection accuracy, due to limitations of the high-precision displacement stage and the experimental environment, this manuscript only demonstrates displacement measurement capability at a resolution of 10 nm. However, we

believe this does not represent the fundamental limit of the multimode fiber-based sensing system. Theoretically, speckle fields with super-oscillation characteristics possess the potential for sub-nanometric resolution (Ref: Opt. Lett., 33, 2976-2978, 2008; Ref: APL Photonics, 7, 086103, 2022).

[editorial note: third party material redacted]

Fig. R13. Comparison of setups for multimode fiber speckle displacement detection and white light interferometer. **a**, Integrated multimode fiber optic detection system. It has a simple structure and low cost, and can be inserted into narrow spaces to detect targets. **b**, Structure and Measurement Principle of the White Light Interferometer System. (Ref: Sensors, 23, 8307, 2023) The optical path is complex and costly, requiring the sample to be exposed at the surface.

Table R1. Parameters of commercial and literature-reported white light interferometers.

No.	Products/Citations	Axial resolution	Lateral resolution	Detection range	Camera effective pixels	Light source
Bruker Contour X-1000	<0.01 nm	0.38 μm	10 mm	5 million	Dual LED
Bruker InSight WLI	0.01 nm	0.38 μm	1.5 mm	-	Dual LED
Sensofar S neox	0.1 nm (PSI/ePSI), 0.01 nm (with PZT), 1 nm (VSI)	-	200 μm (PZT) / 40 mm (motor)	1232x1028	Multispectral LED
Mahr MarSurf WI 50/100	0.2 nm (STR, 100X), 0.43 nm (ISO 25178-604, 100X)	0.34 μm (100X)	0.25 mm (PZT) / 70-100 mm (motor)	1.44/5 million	High-pass LED (650 nm)
Pavel Pavliček et al., Optics and Lasers in Engineering 124 (2020) 105800	-	34 μm	0.260 mm	1.05 million	Superluminescent diode
Yuchu Dong et al., Measurement 186 (2021) 110199	-	1.1 μm	100 μm	10 million	White LED

(3) Finally, regarding the concern you raised, we acknowledge that our method indeed requires training for specific objects. This is a common characteristic of all data-driven deep learning approaches. However, we wish to emphasize that this does not inherently limit the general applicability and versatility of the system.

Firstly, we have demonstrated in the manuscript that the proposed displacement recognition network achieves high-accuracy, high-resolution recognition across various detected patterns and material types. Secondly, to address training and data acquisition time, we significantly increase the speed of speckle data collection by reducing the region of interest, allowing the camera to operate at a lower resolution over a smaller area. For instance, within a 128x128 pixel area, a standard industrial camera can achieve an acquisition rate of 200 Hz. This means that data acquisition at a single displacement position requires only 2.5 seconds. For a sparse sampling task involving 20 points, and accounting for the movement time of the displacement stage, the total time required to collect the entire training dataset is less than one minute. The use of a higher-performance high-speed camera could theoretically reduce this acquisition time by an order of magnitude or more.

This efficiency ensures that even if re-acquisition and re-training are necessary when changing samples, rapid recalibration of the recognition network remains feasible. Furthermore, in the revised manuscript, we also demonstrate the powerful generalization capability of the image reconstruction network across different types of datasets.

Finally, we would like to demonstrate that the method proposed in this manuscript offers distinct advantages and innovations in the following aspects:

(1) A novel high-accuracy in situ nano-displacement detection scheme and configuration are proposed.

(2) Deep learning is introduced into the metrology, with joint learning enabling the detection of various structured targets and materials for the first time.

(3) Superoscillatory in the multimode fiber speckle field is revealed, demonstrating the nano-metrology capability of this strongly scattering media.

(4) The submillimeter, flexible fiber probe is capable of simultaneously performing both positioning and imaging tasks.

The following table shows some representative relevant articles in this field for comparison. It is evident that this manuscript has an irreplaceable performance in terms of probe size, high resolution and accuracy, and single-ended non-contact flexible detecting structure.

Table R2. Comparisons of this manuscript with previous similar publications.

No.	Citations	Principle	System Indicators			Integrated structure	Single-ended detection	In situ	Annotation
			Diameter of Probe	Resolution of Positioning	Accuracy of Positioning				
1	A. Snigirev, Physical Review Letters, 103, 064801 (2009)	Interferometer	Several millimeters	Tens of nanometers	-	✗	✓	✗	Environmentally sensitive and inflexible
2	S. Fericean, IEEE Sensors Journal, 7, 1538-1545 (2007).	Eddy Current	30 mm	14 μm	>97%	✓	✓	✗	Large probe size, inaccessible
3	L. Viry, Advanced Materials, 26, 2659-2664 (2014).	Capacitive Sensing	8 mm	8 μm	-	✓	✓	✗	
4	G. Yuan, Science, 364, 771-775 (2019).	Meta-surface	$\sim 40 \mu\text{m}^*$	<1 nm	-	✓	✗	✗	High optical parameter requirements, difficult to fabrication
5	H. Zang, Science Advances, 10, eadk2265 (2024)		900 μm^*	0.3 nm	error < 1 Å	✓	✗	✗	
6	C. Chen, Nature Communications, 15, 8391 (2024).	Orbital Meta-atoms	-	1.2 μm	error < 0.2%	✗	✗	✗	Bilateral access target
7	J. Xu, Nature Nanotechnology, 19, 1472-1477 (2024).	Single-cavity Loss	-	2 nm	-	✗	✗	✗	
8	This Work	Super-oscillatory Speckle + Deep Learning	800 μm	10 nm	>99.95%	✓	✓	✓	Single-ended in-situ highly robust probing

Note:
*: Size of the metasurfaces

Comment 1: As shown in Figure S4a, the VGG displacement recognition network is a classification network with an output layer of five neurons corresponding to 0 nm, 10 nm, 20 nm, 30 nm, and 40 nm. This method can only predict discrete displacement values (10 nm resolution). To achieve continuous displacement measurement with higher resolution, is it feasible to develop a neural network that directly outputs specific displacement values?

Response:

We thank you for this insightful and important question. To enable higher-resolution prediction, we reformulated the original **discrete classification problem into a regression task**, allowing the network to output continuous displacement values rather than being restricted to fixed class labels. To achieve this, we propose two strategies: Gaussian Label Regression and encoded rectangular patterns.

Firstly, we adopted a method known as **Gaussian Label Regression**, where the conventional one-hot labels are replaced with soft Gaussian-distributed labels centered on the true displacement. This introduces inter-class continuity and allows the model to learn a smoother and more generalizable output space.

During training, we employed a hybrid loss function. The KL divergence measures the difference between the predicted probability distribution and the Gaussian-shaped target, while the mean squared error (MSE) penalizes the difference between the predicted expected displacement and the ground-truth value. As shown in Fig. R14 this strategy enables the network to produce continuous outputs.

Fig. R14 Illustration of the Gaussian soft-label network. The network employs Gaussian labels centered at the true displacement, assigning decreasing probabilities to neighboring classes instead of conventional one-hot labels. A combined KL divergence and MSE loss function is used during training. Even-indexed classes are marked as the training set, while odd-indexed classes serve as the test set.

To evaluate the method’s generalization capability, we divided the displacement classes by parity: even-indexed classes (0, 20, 40 nm...) were used for training, a small portion of odd-indexed samples (10, 30 nm...) were included for validation, and the remaining odd classes were used as the test set. As shown in Fig. R15, the model accurately predicted displacements even for the test classes not seen during training confirming the method’s effectiveness. The prediction accuracy for even classes is relatively high, while that for odd classes is lower. There remains room for improving the choice of Gaussian hyperparameters and loss functions in this method.

Fig. R15 Displacement recognition results of the Gaussian soft-label network.

Fig. R16 Schematic diagram of the U-Net architecture. The network repeatedly uses two consecutive blocks of “Convolution + Batch Normalization (BN) + ReLU”, which is a typical structure in U-Net. Skip connections (indicated by green arrows in the left panel) are incorporated to better retain features across different levels. The right panel illustrates the S-shaped encoding scheme of the input speckle patterns and the decoding process based on the centroid position of the predicted output pattern.

The second method **encodes speckle patterns corresponding to different displacements with rectangular patterns** at varying positions and trains a U-Net network, as illustrated in Fig. R16. Specifically, speckle patterns at each displacement are mapped to rectangular patterns at different locations, where the rectangle’s position follows an S-shaped trajectory as displacement increases. After training, the network takes a speckle pattern as input and outputs

the corresponding pattern. By calculating the centroid of the output pattern and decoding its position according to the encoding scheme, a continuous displacement prediction can be obtained, enabling precise continuous distance measurement.

Fig. R17 shows a set of experimental results using the same data splitting strategy as the first method: even-indexed classes serve as the main training set, a small portion of odd-indexed classes are mixed into the validation set, and the remaining odd-indexed classes are used as the test set. The results demonstrate good performance on the test set, achieving accurate continuous displacement prediction.

Fig. R17 Displacement recognition results of the U-Net network.

The prediction accuracy for even classes (Test A) is relatively high, while that for odd classes (Test B) is lower. There remains significant room for improving the choice of Gaussian hyperparameters and loss functions in this method.

To demonstrate the continuous displacement value recognition capability of the proposed multimode fiber in-situ displacement metrology solution within a deep learning framework, and to validate two feasible implementation approaches for networks directly outputting specific displacement values, we have incorporated relevant content, experimental details, and results into the discussion section of the manuscript and the Supplementary Information. The specific modifications are as follows:

(From page 14, line 429)

resolution of the multimode fiber displacement measurement system. Although the preceding
 discussion only demonstrated discrete displacement recognition results obtained using classified-
 like networks, neural networks based on superoscillatory speckle can output continuous
 displacement values through strategies such as Gaussian Label Regression and encoded rectangular
 patterns. Detailed methods and experimental results are presented in Supplementary Note 5 and
 Supplementary Fig. S11–S14.^e

(Supplementary Information, from page 11, line 382)

**Supplementary Note 5. Continuous recognition capability of the system** ↵
↵
Although the main text only demonstrated discrete displacement recognition results obtained using
classified-like networks, neural networks based on superoscillatory speckle can output continuous
displacement values. To achieve higher resolution and continuous displacement measurement, we
reformulated the original discrete classification problem into a regression task, allowing the
network to output continuous displacement values rather than being restricted to fixed class labels.
To achieve this, we propose two strategies here: Gaussian Label Regression and encoded
rectangular patterns. ↵
↵
Firstly, we adopted a method known as Gaussian Label Regression, where the conventional one-
hot labels are replaced with soft Gaussian-distributed labels centered on the true displacement.
This introduces inter-class continuity and allows the model to learn a smoother and more
generalizable output space. ↵
↵
During training, we employed a hybrid loss function. The KL divergence measures the difference
between the predicted probability distribution and the Gaussian-shaped target, while the mean
squared error (MSE) penalizes the difference between the predicted expected displacement and
the ground-truth value. As shown in Fig. S11 this strategy enables the network to produce
continuous outputs. ↵
↵
To evaluate the method's generalization capability, we divided the displacement classes by parity:
even-indexed classes (0, 20, 40 nm...) were used for training, a small portion of odd-indexed
samples (10, 30 nm...) were included for validation, and the remaining odd classes were used as
the test set. As shown in Fig. S12, the model accurately predicted displacements even for the test
classes not seen during training confirming the method's effectiveness. The prediction accuracy
for even classes is relatively high, while that for odd classes is lower. There remains room for
improving the choice of Gaussian hyperparameters and loss functions in this method. ↵
↵
The second method encodes speckle patterns corresponding to different displacements with
rectangular patterns at varying positions and trains a U-Net network, as illustrated in Fig. S13.
Specifically, speckle patterns at each displacement are mapped to rectangular patterns at different
locations, where the rectangle's position follows an S-shaped trajectory as displacement increases.
After training, the network takes a speckle pattern as input and outputs the corresponding pattern.
By calculating the centroid of the output pattern and decoding its position according to the
encoding scheme, a continuous displacement prediction can be obtained, enabling precise
continuous distance measurement. ↵
↵
Fig. S14 shows a set of experimental results using the same data splitting strategy as the first
method: even-indexed classes serve as the main training set, a small portion of odd-indexed classes
are mixed into the validation set, and the remaining odd-indexed classes are used as the test set.
The results demonstrate good performance on the test set, achieving accurate continuous
displacement prediction. ↵
↵
The prediction accuracy for even classes (Test A) is relatively high, while that for odd classes (Test
B) is lower. There remains significant room for improving the choice of Gaussian hyperparameters
and loss functions in this method. ↵

(Supplementary Information, from page 33, line 989)

(Supplementary Information, from page 34, line 996)

(Supplementary Information, from page 35, line 999)

(Supplementary Information, from page 36, line 1007)

Comment 2: In Figure S18, the 2 mm measurement range is validated at 0.5 mm intervals, which are excessively large. Simultaneously achieving high resolution and long measurement range would require a huge, complex network (e.g., 200,000 neurons in the output layer for 10 nm resolution over 2 mm). Moreover, the volume of speckle data requiring calibration across different displacements is enormous. Are there effective strategies to balance long range and high resolution?

Response:

We sincerely thank you for this insightful and important question. To address this issue, we explored strategies to reduce the data burden and simplify the classification task.

First, it should be clarified that Figure S18 displays image reconstruction results at different displacement positions. The reason for not using high displacement resolution here is that imaging quality remains almost unaffected under nanometer-scale displacement changes.

We fully agree with your observation that achieving high resolution over a large displacement range would require a very large number of neurons in the neural network and a massive amount of speckle data, leading to substantial computational demands. However, several strategies can be employed to balance these requirements, enabling training with small datasets and low computational cost while still achieving high accuracy. Specifically, our approach includes training with small datasets, speckle size compression, and multi-interval partitioned recognition. Each of these strategies is detailed below.

(1) Training with Small Datasets

Fig R18 Confusion matrices obtained from 500 speckles (for training, validation, and testing) of 5 sampling points and 20 sampling points.

To verify the capability of training with limited samples, we selected 500 samples per displacement position at a resolution of 10 nm for training. These 500 samples include the training, validation, and test sets, with the training set accounting for 70%. Thus, only 350 speckle images were used for network training during the model training phase. Figure R18 shows the experimental results using 500 samples for 5 sampling points and 20 sampling points, respectively. The confusion matrices obtained from testing the network with the test set demonstrate that even with such a small dataset, an recognition accuracy of 99.95% can be

achieved for 5 sampling points, and 98.42% for 20 sampling points. Additionally, our approach achieves an accuracy of 95.48% using just 100 speckles per group as the training set. This strategy effectively reduces the memory consumption of network training and significantly decreases both the data volume and computational cost in scenarios involving large displacement ranges and high resolution.

(2) Speckle Size Compression

The second strategy involves cropping and compressing the acquired diffuse speckle patterns. As emphasized in the main text, high-accuracy displacement recognition can be achieved using only a portion of the output speckle field. Fig. R19 displays the confusion matrix when the speckle images are compressed to 8×8 pixels. This result demonstrates that even under such an extreme compression ratio, the network can still achieve very high recognition accuracy. A compression ratio of 0.28‰ significantly reduces the amount of data input to the network, effectively decreasing the number of neurons in the input layer and the computational cost. Speckle compression enables the same computational system to achieve displacement recognition over larger ranges and at higher resolutions.

Fig. R19 Displacement measurement results of speckles under extreme compression ratios. a, Schematic Diagram of Speckle Compression. **b,** Displacement recognition confusion matrix at a compression ratio of 0.28‰.

(3) Multi-interval Partitioned Recognition

Furthermore, we propose a hierarchical classification strategy to effectively balance the resolution and range. Specifically, we first perform coarse classification by grouping adjacent displacements (e.g., 10 neighboring classes) into a super-class to locate the rough displacement interval, followed by fine classification within the identified interval to achieve high-resolution discrimination. The overall process concept is shown in Fig. R20. This two-stage approach significantly reduces the output dimensionality and computational demand while maintaining high accuracy.

Step 1: Coarse classification

Step 2: Fine classification

Fig. R20 Overall process concept of multi-interval partitioned recognition.

Fig. 21 Hierarchical classification strategy for long-range and high-resolution displacement recognition. a, Confusion matrix of the coarse classification, where every 5 adjacent displacement classes are grouped into one super-class. The model effectively distinguishes between these coarse intervals, demonstrating the feasibility of using hierarchical classification to extend the measurable displacement range. **b**, Confusion matrices from two representative fine classification tasks within selected intervals. In both cases, the model accurately resolves fine-grained displacement variations, indicating stable performance and high spatial resolution in local regions.

As a demonstration, we conducted additional experiments and collected experimental data

over a large range of 0–500 nm at a high resolution of 10 nm, totaling 50 displacement sampling points. Following the above strategy, we first performed coarse classification by grouping every five adjacent displacement points into one category. After completing this initial step, the corresponding interval was identified, and a second displacement recognition network was trained using fine-classification data to conduct detailed classification. Fig. R21 presents the confusion matrices for the coarse classification (each super-category containing five classes) and two corresponding fine-classification tasks. As shown in Fig. R21, these results collectively validate the effectiveness of the proposed hierarchical strategy.

In scenarios requiring even higher resolution and larger ranges, the number of hierarchical levels can be further increased to implement multi-stage recognition. This approach compresses the number of neurons in the network and reduces computational cost at the expense of certain time overhead, thereby achieving high-accuracy displacement recognition over extensive ranges and at high resolutions.

The Supplementary information has already detailed the two strategies of Training with Small Datasets and Speckle Size Compression, demonstrating their effectiveness at high resolutions across large ranges. In Note 4 and Fig. S10 of the supplementary materials, we have added an introduction and results for the Multi-interval Partitioned Recognition strategy. This demonstrates the potential for further expansion and enhancement of the proposed methods. The specific modifications are as follows:

(Supplementary Information, from page 10, line 353)

In addition to the two strategies mentioned above (Training with Small Datasets and Speckle Size
Compression) we can also achieve high-resolution measurements across a wide range using the
Multi-interval Partitioned Recognition method. This approach does not require large, complex
networks, nor does it necessitate increasing the number of neurons or computational costs.
←
The hierarchical classification strategy to effectively balance the resolution and range. Specifically,
we first perform coarse classification by grouping adjacent displacements (e.g., 10 neighboring
classes) into a super-class to locate the rough displacement interval, followed by fine classification
within the identified interval to achieve high-resolution discrimination. The overall process
concept is shown in Fig. S10a. This two-stage approach significantly reduces the output
dimensionality and computational demand while maintaining high accuracy.
←
As a demonstration, we conducted additional experiments and collected experimental data over a
large range of 0–500 nm at a high resolution of 10 nm, totaling 50 displacement sampling points.
Following the above strategy, we first performed coarse classification by grouping every five
adjacent displacement points into one category. After completing this initial step, the
corresponding interval was identified, and a second displacement recognition network was trained
using fine-classification data to conduct detailed classification. Fig. S10b presents the confusion
matrices for the coarse classification (each super-category containing five classes) and two
corresponding fine-classification tasks. As shown in Fig. S10c, these results collectively validate
the effectiveness of the proposed hierarchical strategy.
←
In scenarios requiring even higher resolution and larger ranges, the number of hierarchical levels
can be further increased to implement multi-stage recognition. This approach compresses the
number of neurons in the network and reduces computational cost at the expense of certain time
overhead, thereby achieving high-accuracy displacement recognition over extensive ranges and at
high resolutions.

 **Fig. S10. Multi-interval partitioned recognition strategy.** **a**, Overall process concept of multi-
 interval partitioned recognition. **b**, Hierarchical classification strategy for long-range and high-
 resolution displacement recognition. Confusion matrix of the coarse classification, where every 5-
 adjacent displacement classes are grouped into one super-class. The model effectively
 distinguishes between these coarse intervals, demonstrating the feasibility of using hierarchical
 classification to extend the measurable displacement range. **c**, Confusion matrices from two
 representative fine classification tasks within selected intervals. In both cases, the model accurately
 resolves fine-grained displacement variations, indicating stable performance and high spatial
 resolution in local regions.

Comment 3: In Figure 4f, displacement measurement and material identification achieve 100% accuracy at 100 nm resolution, but the intervals are large. Will increasing the number of neurons in the output layer (for higher resolution or longer range) lead to a decrease in accuracy?

Response:

We sincerely thank you for this valuable comment. At a resolution of 100 nm, the displacement accuracy for different metallic materials reached 100%. In the main text, using a wafer as the detected target, we achieved a recognition accuracy of 98.42% at a resolution of 10 nm across a broader measurement range (0–200 nm, 20 sampling points).

To further investigate the impact of increasing the number of neurons in the output layer on classification performance, we conducted an additional experiment with a finer displacement resolution. Specifically, we expanded the displacement measurement range of 0–990 nm into 100 classes at 10 nm intervals, corresponding to 100 neurons in the output layer. The resulting confusion matrix is shown in Fig. R22. Even under such a large number of output neurons, the model still achieved a high classification accuracy of 97.8%, demonstrating the robustness of our approach to higher-resolution displacement classification.

Fig. R22 Confusion Matrix for Displacement Recognition at 10nm High Resolution with a Wide Measurement Range of 0–990 nm (100 Sampling Points).

As seen from the above analysis, increasing the number of neurons in the output layer does indeed lead to a decline in accuracy. However, even under high-resolution (10 nm) and wide-range (0–990 nm) tasks, the proposed method still achieves an accuracy rate of 97.8%. Additionally, data processing strategies such as Multi-interval Partitioned Recognition can be employed to mitigate the accuracy degradation caused by an excessive number of neurons.

We have incorporated these results from the broader measurement range into the discussion section of the manuscript and provided a confusion matrix diagram in the supplementary materials:

(From page 14, line 422)

In terms of measurement range, we have validated the system's detection ability within a 2 mm
range. We performed displacement recognition at 0.5 mm intervals, covering a range from 0-2 mm
(see Supplementary Fig.S8 and S28). However, due to computational constraints, we carried out
recognition for 100 displacement groups at 10 nm resolution (Average accuracy of 97.8%, see Fig.
S9 in the Supplementary Information). With more advanced computational units and larger memory
capacity, the system can achieve a larger measurement range at nano-resolution. Regarding

(Supplementary Information, from page 10, line 345)

using just 100 speckles per group as the training set. Fig. S8d and S8e presents the probability
statistics of each sampled group under the correct displacement label with 500 datasets and 100
datasets. Furthermore, we conducted high-resolution (10 nm) testing across the broad displacement
range of 0-990 nm. The confusion matrix, shown in Fig. S9, still achieves an accuracy of 97.8%,
demonstrating the robustness of our approach to larger-range displacement classification.

(Supplementary Information, from page 31, line 974)

Comment 4: The influence of the distance between the speckle detection plane (CMOS sensor) and the fiber output face should be analyzed.

Response:

We thank for your suggestion regarding the completeness of our experimental details. We fully agree with your perspective and have conducted additional displacement recognition experiments under varying distances between the speckle detection plane and the fiber output end face. These experiments were accompanied by an analysis of their impact on displacement recognition accuracy, thereby effectively enhancing the completeness of the manuscript.

In the Supplementary Information of the original manuscript, we provided an analysis of the effect of the distance between the CMOS sensor plane and the fiber output end face on imaging quality. This was intended to help identify the optimal detector reception distance and to provide readers with a more intuitive understanding of the experimental system configuration. While we initially believed that this distance had a negligible effect on displacement recognition and thus focused our discussion on its impact on imaging quality, we acknowledge that, from the perspective of manuscript comprehensiveness, it is necessary to include the relevant analysis.

Specifically, we performed displacement recognition experiments using the same wafer pattern at four different distances from the CMOS sensor: 4.8 mm, 5.8 mm, 6.8 mm, and 7.8 mm. At each distance, we collected 5 sets of diffuse speckle patterns at 10 nm displacement intervals, with each set containing 1000 speckle fields, resulting in a total of 20 sets of data. Each dataset corresponding to a specific distance was independently used for training and testing, yielding the four confusion matrices shown in Fig. R23.

Fig. R23 Displacement recognition confusion matrices at different acquisition distances. Confusion matrices trained at distances of **a**, 4.8 mm, **b**, 5.8 mm, **c**, 6.8 mm, and **d**, 7.8 mm from the CMOS sensor.

As can be observed from the four confusion matrices, high-resolution displacement recognition at 10 nm with 100% accuracy was achieved across all tested reception distances. This demonstrates the powerful displacement recognition capability and high robustness of the in-situ measurement system and method proposed in the manuscript.

It should be noted, however, that placing the CMOS sensor too close or too far from the fiber output end may lead to issues such as decreased signal-to-noise ratio or speckle overlap

and over-saturation, which could potentially reduce recognition accuracy. Therefore, parameters such as the dynamic range and photosensitive unit size of the camera should also be considered during system configuration. Nevertheless, we can conclude that, compared to image reconstruction, displacement recognition is largely insensitive to the distance between the CMOS sensor and the fiber output end face, enabling high-accuracy recognition across a broad range of acquiring distances.

Following your suggestion, we have added an analysis of the impact of the distance between the CMOS sensor plane and the fiber output end face on displacement recognition in Note 8.3 of the Supplementary Information. Additionally, we have updated and improved Fig. S14 (Fig. S23 in revised Supplementary Information). The specific additions and the modified figure are presented below.

(Supplementary Information, from page 19, line 750)

powers are 5mW, 6mW, 7.5mW, and 8mW. The results are shown in Fig. S23. It can be seen from
the results that as the air thickness increases, the imaging quality deteriorates. We also performed
displacement recognition experiments using the same wafer pattern at four different distances from
the CMOS sensor: 4.8 mm, 5.8 mm, 6.8 mm, and 7.8 mm. At each distance, we collected 5 sets of
diffuse speckle patterns at 10 nm displacement intervals, with each set containing 1000 speckle
fields, resulting in a total of 20 sets of data. Each dataset corresponding to a specific distance was
independently used for training and testing, yielding the four confusion matrices shown in Fig.
S23. The displacement measurement results were not affected by the measurement distance.
However, if the thickness is reduced to a certain value, the imaging quality may also be degraded
due to the relationship between the speckle size and the sensor pixel size.↵

(Supplementary Information, from page 45, line 1077)

[editorial note: third party material redacted]

Comment 5: The system trains a dual network on 4,000 images. After training, can this network accurately detect displacements and reconstruct images for objects beyond these 4,000 images?

Response:

We thank you for raising the question regarding the actual testing capability of the trained network. During the training process, the data were divided into training, validation, and test sets, with no overlap among them. All results and conclusions presented in the Results section of the manuscript were obtained using the test set after the completion of network training, including both displacement recognition and image reconstruction outcomes. The mentioned dataset of 4000 samples includes training, validation, and test sets, with a ratio of 70%, 10%, and 20%, respectively. The displacement measurement results shown in Fig. 3–5 and the imaging results in Fig. 6 were all derived from the final 20% test set.

[editorial note: third party material redacted]

Fig. R24 Generalization test results obtained using the network trained on the ImageNet dataset.

Diffuse speckle field and reconstruction results for grayscale images of handwritten digits and letters from MNIST and EMNIST datasets.

[editorial note: third party material redacted]

Fig. R25 Generalization Test Results. Generalization tests were conducted on various pattern types, including handwritten digits and letters, Chinese characters, simple geometric patterns, and architectural models. The inverse transfer matrix network demonstrated superior generalization capabilities compared to other convolutional networks.

Furthermore, our additional experiments demonstrate that the network exhibits strong transfer learning ability and is not limited to the types of data used during training. In Fig. R24, we present test results using handwritten digits and simple geometric patterns after training on ImageNet. It is worth highlighting that we also conducted experiments using a simple handwritten digit dataset for training and tested the network with self-designed Chinese characters and geometric patterns. The results, as shown in Fig. R25, also demonstrate robust image reconstruction performance.

The above data all demonstrate the actual testing capability of the trained network. We emphasized this point in the main text and supplemented it with relevant reconstruction results in Fig. 6, as follows:

(From page 12, line 393)

[editorial note: third party material redacted]

Comment 6: The scheme's general applicability is limited, as detecting arbitrary objects would require collecting vast amounts of speckle data for different materials.

Response:

Thank you for your question regarding the general applicability of the proposed method. We acknowledge that our approach indeed requires training for specific objects—a characteristic

shared by all data-driven deep learning methods. However, we would like to emphasize that this does not inherently limit the general applicability and versatility of the system.

Firstly, we have demonstrated in the manuscript that the proposed displacement recognition network achieves high-accuracy, high-resolution recognition across various probe patterns and material types. Secondly, to mitigate the time required for training and data acquisition, we significantly increase speckle data collection speed by reducing the region of interest, allowing the camera to operate at lower resolution over a smaller area. For instance, within a 128×128 pixel region, a standard industrial camera can achieve an acquisition rate of 200 Hz. This means that data acquisition at a single displacement position requires only 2.5 seconds. For a sparse sampling task involving 20 points, and accounting for the movement time of the displacement stage, the total time required to collect the entire training dataset is less than one minute. The use of a higher-performance high-speed camera could theoretically reduce the acquisition time by an order of magnitude or more.

Fig. R26 Procedure for detecting the unknown material of a target.

This efficiency ensures that even when sample replacement necessitates re-acquisition and retraining, rapid recalibration of the recognition network remains feasible. It is worth noting that other interferometric measurement techniques, such as white-light interferometry, also require preliminary calibration for different samples. Therefore, we believe that by shortening and controlling data acquisition and training time, rapid calibration for arbitrary objects can be achieved prior to detection, ensuring the general applicability of the method. The specific implementation process is shown in Fig. R26. Furthermore, in the revised manuscript, we have also demonstrated the powerful generalization capability of the image reconstruction network across diverse dataset types.

To emphasize our method's capability for rapid data acquisition and network correction on untrained materials, we have supplemented the main text with experimental details such as data

acquisition speed.

(From page 12, line 364)

minute and 22 seconds. After 10 epochs, the loss function converges. If the pixel size is compressed
to 4×4 , the total training time (including data preprocessing) can be reduced to 9 minutes. In terms
of data acquisition speed, we significantly increase speckle data collection speed by reducing the
region of interest. For instance, within a 128×128 -pixel region, a standard industrial camera can
achieve an acquisition rate of 200 Hz. This means that data acquisition at a single displacement
position requires only 2.5 seconds. For a sparse sampling task involving 20 points, and accounting
for the movement time of the displacement stage, the total time required to collect the entire training
dataset is less than one minute. This rapid acquisition speed ensures that even when the target
material is not included in the trained dataset, the network can be swiftly corrected.[↵]

Comment 7: The triangular and square markers in Figure 2h are mislabeled.

Response:

Thank you for pointing out the issue with the image labeling. In the revised manuscript, we have corrected this error. The triangular markers and their corresponding fitted curve now represent the variation in the energy proportion of the first 100 LP modes with displacement, while the red square markers and their corresponding fitted curve represent the variation in the energy proportion of the first 1000 LP modes with displacement. Therefore, at corresponding displacement values, the red markers and curve exhibit higher numerical values. In addition to updating the markers in the legend, we have also revised the relevant descriptions in the figure caption accordingly.

It is important to emphasize that in the revised manuscript, we have replaced the target image used in the simulations to enhance the overall coherence and consistency of the article. We have also added simulation results for smaller displacement ranges and higher resolutions. The corresponding textual descriptions in the revised manuscript have been modified accordingly. The updated relevant content is as follows:

(From page 5, line 165)

**Fig. 2** Physical model of MMF and simulation results at different displacements. a, Different
combinations of transmission modes of MMFs are coupled at different detected displacement z , forming
different speckles. b-e, The detected target, fiber input facet, field at the core area, and the recombination
result of the first 1000 modes. f, The first 1000 modes are divided into 10 groups in sequential order,
with each group containing 100 modes. The intensity percentage of each group is the sum of the intensity
percentages of the 100 modes in the group. The intensity percentage of these ten groups and the
combination field at 0 mm, 10 mm, 20 mm, and 30 mm displacements. g, Percentage of the first 20
modes and the distribution of the main modes at the detected displacement of 0 nm. h, Normalized
average mode order (blue left axis), percentage of the first 100 modes (orange right axis) and the first
1000 modes (red right axis) at different detected displacement (0-990nm, 100 sampling points).

Response to Reviewer 3

The authors present a non-contact longitudinal nano-displacement metrology system utilizing multimode fibers (MMF) and deep learning framework. Key innovations include: 1) a physical model correlating input images, MMF mode compositions, and output speckle patterns; 2) multitask processing validity for simultaneous displacement measurement (with 99.95% accuracy and 10 nm resolution) and target imaging; 3) a compact and highly integrable lensless single-end MMF detection system with applications in semiconductor industry and super-resolution endoscopy.

While the manuscript demonstrates compelling results, the following issues require clarification before recommending publication in Nature Communications:

Major issues:

Comment 1: How would reduced fiber probe diameters (for example to use fiber probe 2 in Fig. 4c) affect the measurement accuracy?

Response:

We appreciate your careful review and attention to detail of the manuscript.

Thank you for your inquiry regarding the fiber size. This is a highly constructive question and suggestion. The dimensions of the fiber probe directly determine the range of application scenarios for the system. A thinner probe can access narrower spaces to perform in-situ displacement measurement. We believe that the fiber size presented in the manuscript can be further reduced without compromising its displacement recognition capability. However, a smaller fiber diameter would lead to a decrease in the number of supported modes, thereby affecting the image reconstruction quality. Below, we will provide a detailed argumentation on this matter.

To investigate the impact of probe diameter on measurement accuracy, we conducted simulation experiments. As the fiber size (i.e., the diameter of the optical field transmission region) decreases, the number of modes that can be supported within the fiber also reduces. This results in an increase in the speckle size and a decrease in the number and density of speckles in the output field, as shown in Fig. R27.

Fig. R27 Speckle size and density for different fiber diameters.

It demonstrates the output speckle fields under different fiber core diameters. Until the V-number reaches the cutoff frequency, the fiber can only support the fundamental mode, at which point the output optical field exhibits only a single speckle.

In the field of multimode fiber image transmission, it is widely accepted that the number of modes a fiber can support is on the same order of magnitude as the maximum resolution (i.e., number of pixels) of the image it can transmit. The output speckle field can be regarded as a superposition of all modes in the multimode fiber; therefore, its complexity should also be on the same order of magnitude as the number of pixels in the speckle field (which can be considered as the number of modes in an orthogonal coordinate system).

Based on the above conclusion, we can equate speckle fields under different compression ratios to those collected and output by fiber probes of different diameters. A commonly used formula for estimating the number of modes in a step-index (SI) multimode fiber is:

$$M = \frac{V^2}{2} = \frac{k_0^2 a^2 (n_1^2 - n_2^2)}{2} = \frac{2\pi^2 a^2 (n_1^2 - n_2^2)}{\lambda^2} \quad (3 - 1)$$

The shorter the wavelength λ , the larger the numerical aperture $NA = \sqrt{n_1^2 - n_2^2}$ and the larger the core radius a , the more transverse modes the fiber can accommodate, and the greater the number of pixels in the equivalent speckle light field pattern.

For the output speckle field of 128×128 pixels used in the article, it can be roughly converted to the output field of a multimode fiber containing approximately 16,384 modes. The multimode fiber with a 250 μm diameter used in the manuscript can support approximately 57,000 modes. Therefore, the data volume was effectively compressed during preprocessing, yet this did not affect the accuracy of displacement measurement. This indicates that achieving displacement recognition at a 10 nm resolution does not require utilizing all possible transmission mode combinations.

Similarly, for Fiber Probe 2 in Fig. 4c, which is a triple-clad multimode fiber (BD-S100/120/360-STN), the corresponding transmission multimode fiber has a diameter of 100 μm . Moreover, due to its numerical aperture (NA) of 0.22, which is smaller than the NA of 0.46 of the fiber used in the manuscript, the number of modes it can support at 1064 nm is only about 1,900. Nevertheless, its output speckle field still corresponds to a speckle pattern with a resolution of 43×43 pixels. We input the speckle field at this compression ratio into the displacement recognition network for training. Fig. R28a shows the output results, and as can be seen from the confusion matrix, it still achieves nearly 100% displacement recognition accuracy at a 10 nm resolution.

Fig. R28b demonstrates the displacement recognition capability at a 508 nm resolution for equivalent output fields of multimode fibers with different sizes. Specifically, we tested the displacement recognition performance of four fibers with core diameters of 20 μm , 10 μm , 7.5 μm , and 5 μm . Their output speckle fields were equivalently compressed to resolutions of 8×8, 4×4, 3×3, and 2×2 pixels, respectively. The results indicate that even a 20/100 μm multimode fiber can achieve 100% displacement recognition accuracy. Therefore, we can reasonably infer that multimode fibers with diameters above 20 μm can achieve nearly 100% accurate displacement recognition.

However, further reducing the fiber size would result in an insufficient number of transverse modes, causing the information introduced by the optical field under different

displacements to be inadequately captured by the speckle field, thereby reducing recognition accuracy. Despite this, the above simulation experiments demonstrate the significant potential for further miniaturization of fiber probes. This further confirms the extremely high information density of multimode fibers, suggesting that smaller-diameter fibers could potentially enable measurement capabilities in even more extreme environments in the future.

Fig. R28 Simulation results of displacement recognition capability for fibers of different sizes. a, Microscopic image of fiber probe No. 2 and displacement recognition confusion matrix at 10 nm resolution. **b,** Equivalent speckle field and displacement recognition confusion matrix at 508 nm resolution for fibers with four different core diameters: 20 μm, 10 μm, 7.5 μm, and 5 μm.

On the other hand, compared to displacement recognition, image transmission imposes higher requirements on the number of modes the fiber can support. Specifically, under the same system architecture, we repeated the experiments using the thinner Fiber Probe 2 with a smaller fiber-end-ball in Fig. 4c, and compared the results across different transmission diameters. In this proof-of-concept experiment, the imaging system with a 250 μm fiber equipped with an 800 μm microsphere could probe a circular area approximately 400 μm in diameter, while the system with a 100 μm fiber equipped with a 600 μm microsphere could probe a circular area approximately 260 μm in diameter.

As shown in Fig. R29a, the imaging results from the 250 μm fiber system are clearer than those from the 100 μm fiber system, with better reconstruction of grayscale and details. Since the number of pixels in the images recovered by the two systems differs, the Peak Signal-to-

Noise Ratio (PSNR) was used instead of the Structural Similarity Index Measure (SSIM) to evaluate imaging quality. The average PSNR values for the reconstruction patterns were 18.62 and 17.59 for the 250 μm and 100 μm fiber systems, respectively. Fig. R29b shows the changes in PSNR and Mean Square Error (MSE) during the training process based on the validation dataset. The training process used PSNR as a callback function, with optimized results oscillating upward and converging after approximately 100 iterations. Fig. R29c compares the output speckle fields of the two systems for the same input image. Although reducing the fiber diameter leads to a decrease in the field of view and resolution, the above results demonstrate that the 100 μm fiber system can still image natural scenes with substantial information retention.

[editorial note: third party material redacted]

Fig. R29 Comparison of imaging quality between Probe No. 1 (250 μm) and Probe No. 2 (100 μm). **a**, Pictures of natural scenes and the corresponding reconstruction patterns which imaging by MMF with 100 and 250 μm core respectively. **b**, Training processes of two diameter systems. 2D-correlation for the validation data sets. **c**, The output speckle images of the same input pattern captured by two diameter systems.

Comment 2: It will be more fundamental to study the dependence of displacement measurement on the surface roughness. If the roughness is comparable or even above the minimum resolution (10 nm for example), is the deep learning algorithm still applicable and reliable?

Response:

We appreciate your attention to the surface roughness of the samples. This is an interesting and critical question. As you pointed out, studying the dependence of displacement measurement on the sample is important, as it determines the applicable scenarios and generalizability of the measurement method. Surface roughness is one of the key parameters characterizing the sample for displacement measurement.

First, it should be noted that the method proposed in this manuscript, which employs diffuse speckle fields and a displacement recognition network, is not strongly dependent on the surface roughness of the sample. In other words, the method is applicable to samples with varying degrees of roughness, and within a wide range, surface roughness does not significantly affect the displacement recognition results. The following provides a more detailed analysis.

Except for certain mirror-like and engineered materials that undergo high-precision polishing techniques such as magnetorheological finishing or ion-beam figuring, the surface roughness of real-world objects is generally larger than 10 nm (the minimum displacement resolution presented in the manuscript). The materials used in our manuscript, such as wafers and metal plates, fall into this category.

To verify the surface roughness of the test samples, we measured a wafer and four additional metal plates (stainless steel, brass, aluminum, and copper) using a Park Systems atomic force microscope (AFM). Fig. R30 shows the selected sampling areas on the wafer surface and the measured height distribution within a $5\ \mu\text{m} \times 5\ \mu\text{m}$ region. Figure R31 further presents the two-dimensional height distributions and height histograms of the selected areas measured by AFM for all five material samples, illustrating the surface roughness.

Fig. R30 Surface roughness of the wafer measured by atomic force microscopy (AFM). **a**, Selected sampling area on the wafer surface. **b**, Height distribution within the measured $5\ \mu\text{m} \times 5\ \mu\text{m}$ region.

It can be observed that within the selected sampling area of the wafer, the height variation, i.e., the peak-to-valley (PV) value, is approximately 50 nm. The other four metal plates (stainless steel, brass, aluminum, and copper) exhibit higher roughness, with PV values of 293 nm, 165.5 nm, 163 nm, and 150.5 nm, respectively. From their respective histograms and fitted curves, the height variations approximately follow a Gaussian distribution centered around zero. The above results are based on the height distributions measured within a $5\ \mu\text{m} \times 5\ \mu\text{m}$

sampling region. In contrast, the fiber probe used in the manuscript detects a larger area of approximately $400\ \mu\text{m} \times 400\ \mu\text{m}$, within which the samples are expected to exhibit even larger PV values and roughness. Even so, the proposed method, based on diffuse speckle fields and deep learning recognition, can still achieve nearly 100% recognition accuracy for the wafer at 10 nm resolution and for the metal materials at 100 nm resolution. This demonstrates the robustness and general applicability of our method to samples with varying surface roughness.

We would like to further emphasize that the method proposed in the manuscript is aimed at relative displacement measurement of the target object. It does not involve measuring the absolute height variations or absolute distances within the detection area (although this could be achieved through calibration if desired). For a rough surface, even if its height variations exceed the minimum measurement resolution, when the sample undergoes a displacement, the displacement at each point within the detection area is identical. Therefore, within a certain range, as long as the surface roughness does not significantly affect the reflected signal intensity, the height variations of the sample do not impact the measurement accuracy.

Fig. R31 Two-dimensional surface roughness distributions and height histograms of five different material samples. **a**, Wafer surface. **b**, Stainless steel plate. **c**, Brass plate. **d**, Aluminum plate. **e**, Copper plate.

To prevent readers from having similar concerns, we have added a discussion of the sample surface roughness to the Supplementary Materials. The specific revisions are as follows:

(From page 10, line 319)

networks in optical metrology, enabling multitask processing. Detailed experimental information
 can be found in the Supplementary Note 7 and Supplementary Fig.S17. Additionally, it should be
 noted that the wide-field detection scheme imposes no special requirements on the roughness of the
 material being tested. As long as the reflectivity is met, the system can be applied to uneven samples.
 The results and analysis of the roughness measurements for the wafers and metal plates mentioned
 above can be found in the Fig.S18 of Supplementary Information. ↵

(Supplementary Information, from page 14, line 546)

Additionally, surface roughness is one of the key parameters characterizing the sample for
 displacement measurement. First, it should be noted that the method proposed in this manuscript,
 which employs diffuse speckle fields and a displacement recognition network, is not strongly
 dependent on the surface roughness of the sample. In other words, the method is applicable to
 samples with varying degrees of roughness, and within a wide range, surface roughness does not
 significantly affect the displacement recognition results. The following provides a more detailed
 analysis.⁴
 Except for certain mirror-like and engineered materials that undergo high-precision polishing
 techniques such as magnetorheological finishing or ion-beam figuring, the surface roughness of
 real-world objects is generally larger than 10 nm (the minimum displacement resolution presented
 in the manuscript). The materials used in our manuscript, such as wafers and metal plates, fall into
 this category.⁴
 To verify the surface roughness of the test samples, we measured a wafer and four additional metal
 plates (stainless steel, brass, aluminum, and copper) using a Park Systems atomic force microscope
 (AFM). Fig. S18a and S18b shows the selected sampling areas on the wafer surface and the
 measured height distribution within a $5\ \mu\text{m} \times 5\ \mu\text{m}$ region. Fig. S18c-S18g further presents the
 two-dimensional height distributions and height histograms of the selected areas measured by
 AFM for all five material samples, illustrating the surface roughness.⁴
 It can be observed that within the selected sampling area of the wafer, the height variation, i.e., the
 peak-to-valley (PV) value, is approximately 50 nm. The other four metal plates (stainless steel,
 brass, aluminum, and copper) exhibit higher roughness, with PV values of 293 nm, 165.5 nm, 163
 570 nm, and 150.5 nm, respectively. From their respective histograms and fitted curves, the height
 variations approximately follow a Gaussian distribution centered around zero. The above results
 are based on the height distributions measured within a $5\ \mu\text{m} \times 5\ \mu\text{m}$ sampling region. In contrast,
 the fiber probe used in the manuscript detects a larger area of approximately $400\ \mu\text{m} \times 400\ \mu\text{m}$,
 within which the samples are expected to exhibit even larger PV values and roughness. Even so,
 the proposed method, based on diffuse speckle fields and deep learning recognition, can still
 achieve nearly 100% recognition accuracy for the wafer at 10 nm resolution and for the metal
 materials at 100 nm resolution. This demonstrates the robustness and general applicability of our
 method to samples with varying surface roughness.⁴
 We would like to further emphasize that the method proposed in the manuscript is aimed at relative
 displacement measurement of the target object. It does not involve measuring the absolute height
 variations or absolute distances within the detection area (although this could be achieved through
 calibration if desired). For a rough surface, even if its height variations exceed the minimum
 measurement resolution, when the sample undergoes a displacement, the displacement at each
 point within the detection area is identical. Therefore, within a certain range, as long as the surface
 roughness does not significantly affect the reflected signal intensity, the height variations of the
 sample do not impact the measurement accuracy.⁴

(Supplementary Information, from page 40, line 1031)

Comment 3: There is a technical challenge that the fiber must be aligned vertically with respect to the sample surface, otherwise the fields of view will change at different heights. If the fiber is slightly tilted, what will happen in the output speckle fields? How about tolerance on the misalignment angles?

Response:

Thank you for your questions regarding the experimental details. As you mentioned, ideally the fiber probe should be aligned vertically with respect to the sample surface. However, in practical experiments and real-world usage, strict vertical alignment is difficult to achieve. We would like to point out that even if the fiber is not perfectly perpendicular to the sample surface, as long as the displacement direction remains parallel to the fiber axis, the field of view at different heights can still remain consistent. On the other hand, if the displacement direction forms an angle with the fiber, the field of view and the detected speckle pattern will change with height. Nevertheless, as long as this angle remains constant, accurate displacement recognition is still achievable, as demonstrated in Fig. 3e of the manuscript.

To address your concern regarding the fiber-to-sample tilt, we performed comparative experiments to investigate its effect on displacement recognition. It should be noted that most of our samples exhibit specular reflection, meaning that there exists a specific reflection angle at which the returned signal is significantly stronger within a certain range. In the comparative experiment, the wafer was placed on a rotation stage, and the reflected light received by the camera was observed. The midpoint of the high-intensity reflection angle range was taken as 0° . The rotation stage was then sequentially rotated, and the diffuse speckle fields captured by the camera were recorded at $\pm 0.5^\circ$, $\pm 1.5^\circ$, $\pm 2.5^\circ$, $\pm 3.5^\circ$, and $\pm 4.5^\circ$, while keeping the sample-to-fiber distance constant, as shown in Fig. R32. From the figure, it can be seen that as the tilt angle deviates further from 0° , the intensity of the captured returned light gradually decreases, resulting in lower speckle contrast and reduced signal-to-noise ratio. When the tilt angle exceeds 5° , the camera barely receives any returned signal, making effective acquisition impossible.

Fig. R32 Diffuse speckle fields captured at the output when the sample normal and fiber axis are misaligned at different angles.

Furthermore, to directly evaluate the impact of tilt angle on displacement recognition, we acquired five sampling points at each angle with a displacement resolution of 10 nm, and for each sampling point, 1000 diffuse speckle fields were collected. These data were then input

into the displacement recognition network for training. The resulting confusion matrices for each angle are shown in Fig. R33. From the figure, it can be seen that within the range of -1.5° to 0.5° , displacement recognition achieves 100% accuracy. When the probe tilt exceeds this range, the recognition accuracy drops sharply, rendering the results completely unreliable. Therefore, we conclude that the system can achieve high-accuracy displacement recognition within an effective angular range of approximately $\pm 2^\circ$. The fiber does not need to be perfectly perpendicular to the sample surface, but it should remain within this effective angular range.

In practice, to ensure effective data acquisition, before each experimental session, the tilt angle between the sample and fiber is adjusted in real time by observing the intensity of the diffuse speckle fields captured by the output camera. This procedure ensures that the acquired data maintain sufficient signal-to-noise ratio for reliable displacement recognition.

Fig. R33 Confusion matrices for displacement recognition at different tilt angles. The angle between the wafer sample normal and the fiber axis is (a) 0° , (b) 0.5° , (c) 1.5° , (d) 2.5° , (e) -0.5° , (f) -1.5° , and (g) -2.5° , respectively, showing the results obtained from training the displacement recognition network at each tilt angle.

On the other hand, regarding the imaging capability, we conducted a comparison experiment by varying the declination angle of the DMD at an interval of 0.5° each time while keeping the rest parts of the system and the distance between the DMD and the probe fixed. We took a total of 5 positions above and below the nominal 12° rotation angle of the DMD, and collecting 8000 sets of data at each position for the test. The experimental results are shown in Figure R34.

[editorial note: third party material redacted]

Fig. R34 Results of imaging with different declination angles of the DMD.

The declination angle of the DMD within a certain range has little effect on imaging quality. However, if it deviates too far from 12° , the signal light coupled back into the fiber is

significantly reduced, and the signal-to-noise ratio drops sharply, which adversely affects the imaging quality.

To illustrate this experimental detail, we have added the relevant description and experimental results in the Supplementary Materials. The specific modifications are as follows:

(Supplementary Information, from page 19, line 765)

**9.4 Optimization of the detection angle**⁴³
⁴³
Ideally, the fiber probe should be aligned vertically with respect to the sample surface. However,
in practical experiments and real-world usage, strict vertical alignment is difficult to achieve. We
would like to point out that even if the fiber is not perfectly perpendicular to the sample surface,
as long as the displacement direction remains parallel to the fiber axis, the field of view at different
heights can still remain consistent. On the other hand, if the displacement direction forms an angle
with the fiber, the field of view and the detected speckle pattern will change with height.
Nevertheless, as long as this angle remains constant, accurate displacement recognition is still
achievable, as demonstrated in Fig. 3e of the manuscript.⁴³
⁴³
To address your concern regarding the fiber-to-sample tilt, we performed comparative experiments
to investigate its effect on displacement recognition. It should be noted that most of our samples
exhibit specular reflection, meaning that there exists a specific reflection angle at which the
returned signal is significantly stronger within a certain range. In the comparative experiment, the
wafer was placed on a rotation stage, and the reflected light received by the camera was observed.
The midpoint of the high-intensity reflection angle range was taken as 0°. The rotation stage was
then sequentially rotated, and the diffuse speckle fields captured by the camera were recorded at
$\pm 0.5^\circ$, $\pm 1.5^\circ$, $\pm 2.5^\circ$, $\pm 3.5^\circ$, and $\pm 4.5^\circ$, while keeping the sample-to-fiber distance constant, as
shown in Fig. S24a. From the figure, it can be seen that as the tilt angle deviates further from 0°,
the intensity of the captured returned light gradually decreases, resulting in lower speckle contrast
and reduced signal-to-noise ratio. When the tilt angle exceeds 5°, the camera barely receives any
returned signal, making effective acquisition impossible.⁴³
⁴³
Furthermore, to directly evaluate the impact of tilt angle on displacement recognition, we acquired
five sampling points at each angle with a displacement resolution of 10 nm, and for each sampling
point, 1000 diffuse speckle fields were collected. These data were then input into the displacement
recognition network for training. The resulting confusion matrices for each angle are shown in Fig.
S24b. From the figure, it can be seen that within the range of -1.5° to 0.5° , displacement
recognition achieves 100% accuracy. When the probe tilt exceeds this range, the recognition
accuracy drops sharply, rendering the results completely unreliable. Therefore, we conclude that
the system can achieve high-accuracy displacement recognition within an effective angular range
of approximately $\pm 2^\circ$. The fiber does not need to be perfectly perpendicular to the sample surface,
but it should remain within this effective angular range.⁴³
In practice, to ensure effective data acquisition, before each experimental session, the tilt angle
between the sample and fiber is adjusted in real time by observing the intensity of the diffuse
speckle fields captured by the output camera. This procedure ensures that the acquired data
maintain sufficient signal-to-noise ratio for reliable displacement recognition.⁴³

(Supplementary Information, from page 46, line 1083)

 **Fig. S24. Diffuse speckle fields captured at the output and confusion matrices for**
 **displacement recognition at different tilt angles. a,** Diffuse speckle fields captured with the
 **sample normal and fiber axis are misaligned at different angles. b,** The angle between the wafer
 **sample normal and the fiber axis is 0°, ±0.5°, ±1.5°, and ±2.5°, respectively, showing the results**
 **obtained from training the displacement recognition network at each tilt angle.**

Comment 4: The proposed scheme requires a large amount of data for training purposes, and the whole process is quite time-consuming (to take 500 speckle fields per step in several minutes), making it susceptible to environmental vibrations. Any comments on how to balance the accuracy and time?

Response:

We appreciate your concern regarding the time cost of training. As you mentioned, achieving high-precision and complex tasks using neural networks inherently requires a large amount of data. Long acquisition times may make the system susceptible to environmental disturbances. To further reduce the time cost, we propose several strategies: small-data training, speckle size compression, and multiple recognition over partitioned regions.

First, by reducing the region of interest, the camera can capture data at lower resolution over a smaller area, which significantly increases the speckle data acquisition speed. For example, within a 128×128 pixel area, a standard industrial camera can achieve a capture rate of up to 200 Hz. This means that data acquisition at a single displacement position can be completed in only 2.5 seconds. For a sparse task with 20 sampling points, considering the time required for the translation stage to move, the overall time for acquiring the training set is less than 1 minute. Using a higher-performance high-speed camera could theoretically reduce the acquisition time by more than an order of magnitude.

Below, we provide a detailed description of our strategy for balancing accuracy and acquisition time:

(1) Training with Small Datasets

Fig R35 Confusion matrices obtained from 500 speckles (for training, validation, and testing) of 5 sampling points and 20 sampling points.

To verify the capability of training with limited samples, we selected 500 samples per displacement position at a resolution of 10 nm for training. These 500 samples include the training, validation, and test sets, with the training set accounting for 70%. Thus, only 350 speckle images were used for network training during the model training phase. Fig. R35 shows the experimental results using 500 samples for 5 sampling points and 20 sampling points, respectively. The confusion matrices obtained from testing the network with the test set

demonstrate that even with such a small dataset, an recognition accuracy of 99.95% can be achieved for 5 sampling points, and 98.42% for 20 sampling points. Additionally, our approach achieves an accuracy of 95.48% using just 100 speckles per group as the training set. This strategy effectively reduces the memory consumption of network training and significantly decreases both the data volume and time cost in scenarios involving large displacement ranges and high resolution.

(2) Speckle Size Compression

The second strategy involves cropping and compressing the acquired diffuse speckle patterns. As emphasized in the main text, high-accuracy displacement recognition can be achieved using only a portion of the output speckle field. Fig. R36 displays the confusion matrix when the speckle images are compressed to 8×8 pixels. This result demonstrates that even under such an extreme compression ratio, the network can still achieve very high recognition accuracy. A compression ratio of 0.28‰ significantly reduces the amount of data input to the network, effectively decreasing the number of neurons in the input layer and the time cost. Speckle compression enables the same computational system to achieve displacement recognition over larger ranges and at higher resolutions.

Fig. R36 Displacement measurement results of speckles under extreme compression ratios. a, Schematic Diagram of Speckle Compression. **b,** Displacement recognition confusion matrix at a compression ratio of 0.28‰.

(3) Multi-interval Partitioned Recognition

Furthermore, we propose a hierarchical classification strategy to effectively balance the resolution and range. Specifically, we first perform coarse classification by grouping adjacent displacements (e.g., 10 neighboring classes) into a super-class to locate the rough displacement interval, followed by fine classification within the identified interval to achieve high-resolution discrimination. The overall process concept is shown in Fig. R37. This two-stage approach significantly reduces the output dimensionality and computational demand while maintaining high accuracy.

Step 1: Coarse classification

Step 2: Fine classification

Fig. R37 Overall process concept of multi-interval partitioned recognition.

Fig. R38 Hierarchical classification strategy for long-range and high-resolution displacement recognition. **a**, Confusion matrix of the coarse classification, where every 5 adjacent displacement classes are grouped into one super-class. The model effectively distinguishes between these coarse intervals, demonstrating the feasibility of using hierarchical classification to extend the measurable displacement range. **b**, Confusion matrices from two representative fine classification tasks within selected intervals. In both cases, the model accurately resolves fine-grained displacement variations, indicating stable performance and high spatial resolution in local regions.

As a demonstration, we conducted additional experiments and collected experimental data

over a large range of 0–500 nm at a high resolution of 10 nm, totaling 50 displacement sampling points. Following the above strategy, we first performed coarse classification by grouping every five adjacent displacement points into one category. After completing this initial step, the corresponding interval was identified, and a second displacement recognition network was trained using fine-classification data to conduct detailed classification. Fig. R38 presents the confusion matrices for the coarse classification (each super-category containing five classes) and two corresponding fine-classification tasks. As shown in Fig. R38, these results collectively validate the effectiveness of the proposed hierarchical strategy.

In scenarios requiring even higher resolution and larger ranges, the number of hierarchical levels can be further increased to implement multi-stage recognition. This strategy avoids training on excessively large datasets. During practical testing, this approximate binary-search-like approach allows for faster identification of the actual displacement. Consequently, it enables a balance between acquisition time, training time, and high-accuracy displacement recognition at high resolution.

Following your suggestion, we have supplemented the manuscript with additional experimental details regarding training time and data acquisition time. The specific modifications are as follows:

(From page 12, line 364)

to 4×4, the total training time (including data preprocessing) can be reduced to 9 minutes. In terms
of data acquisition speed, we significantly increase speckle data collection speed by reducing the
region of interest. For instance, within a 128×128-pixel region, a standard industrial camera can
achieve an acquisition rate of 200 Hz. This means that data acquisition at a single displacement
position requires only 2.5 seconds. For a sparse sampling task involving 20 points, and accounting
for the movement time of the displacement stage, the total time required to collect the entire training
dataset is less than one minute. This rapid acquisition speed ensures that even when the target
material is not included in the trained dataset, the network can be swiftly corrected.↵

(Supplementary Information, from page 10, line 353)

In addition to the two strategies mentioned above (Training with Small Datasets and Speckle Size
Compression) we can also achieve high-resolution measurements across a wide range using the
Multi-interval Partitioned Recognition method. This approach does not require large, complex
networks, nor does it necessitate increasing the number of neurons or computational costs.↵
↵
The hierarchical classification strategy to effectively balance the resolution and range. Specifically,
we first perform coarse classification by grouping adjacent displacements (e.g., 10 neighboring
classes) into a super-class to locate the rough displacement interval, followed by fine classification
within the identified interval to achieve high-resolution discrimination. The overall process
concept is shown in Fig. S10a. This two-stage approach significantly reduces the output
dimensionality and computational demand while maintaining high accuracy.↵
↵
As a demonstration, we conducted additional experiments and collected experimental data over a
large range of 0–500 nm at a high resolution of 10 nm, totaling 50 displacement sampling points.
Following the above strategy, we first performed coarse classification by grouping every five
adjacent displacement points into one category. After completing this initial step, the
corresponding interval was identified, and a second displacement recognition network was trained
using fine-classification data to conduct detailed classification. Fig. S10b presents the confusion
matrices for the coarse classification (each super-category containing five classes) and two
corresponding fine-classification tasks. As shown in Fig. S10c, these results collectively validate
the effectiveness of the proposed hierarchical strategy.↵
↵
In scenarios requiring even higher resolution and larger ranges, the number of hierarchical levels
can be further increased to implement multi-stage recognition. This approach compresses the
number of neurons in the network and reduces computational cost at the expense of certain time
overhead, thereby achieving high-accuracy displacement recognition over extensive ranges and at
high resolutions.↵

 **Fig. S10. Multi-interval partitioned recognition strategy.** **a**, Overall process concept of multi-
 interval partitioned recognition. **b**, Hierarchical classification strategy for long-range and high-
 resolution displacement recognition. Confusion matrix of the coarse classification, where every 5-
 adjacent displacement classes are grouped into one super-class. The model effectively
 distinguishes between these coarse intervals, demonstrating the feasibility of using hierarchical
 classification to extend the measurable displacement range. **c**, Confusion matrices from two
 representative fine classification tasks within selected intervals. In both cases, the model accurately
 resolves fine-grained displacement variations, indicating stable performance and high spatial
 resolution in local regions.

Comment 5: Speckle field is sensitive to the laser coherence. What's the linewidth and coherence length of the laser source? Is it possible to use the spectrum or wavelength as a new degree of freedom for displacement measurement? A brief discussion will be beneficial.

Response:

We sincerely thank you for this valuable suggestion. The laser used in our manuscript is a narrow-linewidth 1064 nm single-frequency polarization-maintaining fiber laser (Connet, CoSF-D-YB-M-1064-PM-FA) with a linewidth of 20 kHz. Based on Eq. (3–2), the coherence length can be calculated to be approximately 9.95 km in air and 8.20 km in fiber.

$$L = \sqrt{\frac{2 \ln(2)}{\pi n} \frac{\lambda^2}{\Delta\lambda}} \quad (3 - 2)$$

where λ denotes the central wavelength of the source, n represents the refractive index of the medium, and $\Delta\lambda$ corresponds to the spectral linewidth. A narrow-linewidth laser is chosen because it generates output speckle fields with higher contrast. Higher speckle contrast enhances the distinguishability of the output fields under different displacement conditions and for different target images, which facilitates the learning process of both the displacement recognition network and the image reconstruction network.

We will include the detailed light source parameters in the Methods section of the manuscript to provide readers with a clearer understanding of our system configuration, as follows:

(From page 15, line 500)

500	Experimental set-up[†]
501	The laser beam of a 1064 nm wavelength laser (Connet, CoSF-D-YB-M-1064-PM-FA) with a
502	narrow linewidth (20 kHz) is used to illuminate. The MMF used to detect the displacement is a

Regarding your suggestion of using the spectrum or wavelength as a new degree of freedom for displacement measurement, we agree that this is a feasible and novel idea, introducing an additional dimension to displacement sensing. To realize this approach, the system would ideally employ a laser source with a certain spectral bandwidth. However, to ensure sufficient speckle contrast in the output field, the linewidth should not be too broad, so as to maintain a degree of coherence.

When laser light with different wavelengths illuminates the target and returns to the fiber, each wavelength experiences distinct mode distributions, mode-dependent losses, modal dispersion, and intermodal coupling during multimode fiber propagation. By analyzing and processing the spectral information of the output field, it is possible to infer the displacement of the target. If both the two-dimensional speckle pattern and the spectral information of the output light are detected simultaneously, this approach could potentially achieve higher displacement recognition accuracy.

We have added a discussion of the potential for displacement measurement using spectral information in the Discussion section of the manuscript, as follows:

(From page 15, line 466)

466	be realized. Additionally, the system can incorporate new degrees of freedom, such as spectral
467	information, which also influences modal dispersion and inter-mode coupling. The speckle field can
468	restore spectral information while potentially further enhancing displacement recognition accuracy.

Comment 6: In the section of “Image reconstruction with fiber probe”, if the input image is not in the ImageNet dataset, is it still possible to recognize it using the image reconstruction scheme? If yes, add an experimental result in Fig. 6c.

Response:

Thank you for raising concerns regarding the dataset generalization and transfer learning capability of our system in the context of image reconstruction.

In the original manuscript, we demonstrated image reconstruction results using images from ImageNet that were not included in the training set. To further validate the generalization capability of our system and network, we conducted additional experiments. Specifically, we still used a subset of ImageNet images for training, while including Chinese, handwriting digits, handwriting letters, and icon patterns in the test set. The reconstruction results are shown in Fig. R39.

[editorial note: third party material redacted]

Fig. R39 Generalization test results obtained using the network trained on the ImageNet dataset. Diffuse speckle field and reconstruction results for grayscale images of handwritten digits and letters from MNIST and EMNIST datasets.

[editorial note: third party material redacted]

Fig. R40 Generalization Test Results. Generalization tests were conducted on various pattern types, including handwritten digits and letters, Chinese characters, simple geometric patterns, and architectural models. The inverse transfer matrix network demonstrated superior generalization capabilities compared to other convolutional networks.

It can be seen that even when the test images are not part of the ImageNet dataset, our image reconstruction scheme is still able to correctly identify them. This demonstrates that our fiber-probe system and reconstruction network possess strong dataset generalization capability. In fact, in our previous work, we further compared the dataset generalization performance of the inverse transmission matrix approach with that of other convolutional neural networks, and the results revealed its strong transfer learning capability. Specifically, we used a model trained on 50,000 natural scene images to test on other types of images, and the corresponding reconstruction results are shown in Fig. R40.

In Fig. R40, from simple patterns such as digits, letters, and geometric shapes to more complex ones such as Chinese characters and logos, different types of input patterns all achieve satisfactory transfer learning performance. The results of the optimized transmission matrix are shown in the second row of Fig. R40, where high-resolution imaging is realized for all types of patterns.

Following your suggestion, we have supplemented Fig. 6c with additional image reconstruction results on other datasets to further demonstrate the generalization capability. The revised content and the updated Fig. 6 are provided below:

(From page 12, line 393)

[editorial note: third party material redacted]

Comment 7: In Supplementary Note 4, it's mentioned that "Laser powers also need to be adjusted, that the maximum grey value at the center of the speckle is consistent at different positions." Is the laser power adjusted manually or automatically? If this step is mandatory, the calibration process will be very time consuming. For a large displacement range up to a millimeter level, the intensity will change dramatically, how to make sure the calibration reasonable? Will the measurement accuracy be improved if using a large dynamic range CCD, for example 10 bit?

Response:

Thank you for your questions regarding the experimental details. As you noted, in our experiments we deliberately ensured the consistency of the output speckle intensity. This is necessary to prevent overexposure of the speckle pattern at very small displacements or very low signal levels at large displacements, where the signal could be overwhelmed by background light or dark current noise.

The laser power was adjusted manually, not using the camera's automatic exposure function. This is because automatic exposure introduces time delays, requiring a detect-compute-adjust cycle that can take hundreds of milliseconds to seconds per image, which would significantly slow down data acquisition and reduce system robustness and accuracy. In contrast, our manual adjustment is only performed when the intensity appears unreasonable (e.g., overexposure) or changes drastically. For the same displacement, once the exposure is set, it remains fixed. The calibration process is therefore not time-consuming; for example, even if the exposure needs adjustment for each displacement step, calibrating a dataset of 4,000 images takes only about 2 seconds in total, corresponding to an average additional acquisition time of ~0.5 ms per image.

As you mentioned, for large displacement ranges up to the millimeter scale, intensity variations can be significant, making proper calibration critical. Our adjustment criterion is based on the average grayscale value of individual pixels within the region of interest, calculated rapidly by the camera. After exposure adjustment, the average grayscale remains nearly constant across each training task. It is worth noting that this level of consistency is difficult to achieve with the camera's automatic exposure function.

Regarding your suggestion about using a large dynamic range CCD, employing cameras with higher bit depth (e.g., 10-bit or 16-bit) could help alleviate this issue by reducing overexposure and improving the signal-to-noise ratio. However, such cameras are typically expensive. According to our compressed-sampling experiments, the current bit depth does not limit the displacement or imaging precision. Nonetheless, for future systems requiring higher sampling numbers or finer displacement resolution, increasing the camera bit depth and dynamic range is a feasible improvement strategy.

Comment 8: Throughout the manuscript, the authors mainly focused on the longitudinal displacement metrology. How about the transverse resolution? Any qualitative prior knowledge is expected. What could be the ultimate longitudinal and lateral resolution and limiting factors?

Response:

We sincerely appreciate your careful reading and constructive comments. This is an excellent question regarding the extensibility of our system. Due to the limitations of our high-precision displacement testing setup, the present manuscript primarily investigates the longitudinal displacement metrology capability of the multimode fiber probe. This focus is motivated by practical applications such as monitoring vertical movement, vibrations, and surface fluctuations of wafers and similar samples.

Nevertheless, transverse displacement is equally important. In fact, previous studies have already demonstrated the feasibility of transverse displacement metrology using multimode fibers (Ref: *APL Photonics* 7, 086103, 2022). Fig. R41 illustrates the experimental setup. In that work, by exploiting large phase gradients and the phase singularities typically generated in speckle patterns from multimode fibers, the authors experimentally achieved simultaneous two-dimensional displacement resolution of ~ 1.8 nm ($\lambda/300$, where $\lambda=532$ nm is the optical wavelength). This corresponds to a resolution about 670 times smaller than the diffraction limit imposed by the multimode fiber.

[editorial note: third party material redacted]

Fig. R41 Principle of nanoscale measurements with fiber-based optical ruler. Simplified sketch of the experimental setup. (Ref: *APL Photonics* 7, 086103, 2022)

Their experimental results showed that within a 100 nm displacement range, the accuracy reached 1.8 nm (see Fig. R42 of Ref.), while at larger displacements the measured offset gradually deviated from the set value. Furthermore, root-mean-square deviation (RMSD) analysis theoretically demonstrated that when the photon number exceeds 10 (about 5 nW), a 1 nm displacement can be measured with 0.1 nm precision. These findings are highly encouraging and further highlight the strong potential of multimode fibers for displacement metrology.

[editorial note: third party material redacted]

Fig. R42 Experimental and simulation results (Ref: *APL Photonics* 7, 086103, 2022). **a**, Experimentally measured shift of the MMF facet as a function of the displacement set by the piezo stage. **b**, Results of numerical experiments. Root-mean-square deviation (RMSD) as a function of the total number of photons.

We believe that the ultimate longitudinal and transverse resolution is fundamentally determined by the refractive index profile, numerical aperture, and the number of modes supported by the multimode fiber. In practical experimental implementations, however, additional limiting factors include the pixel size, resolution, and bit depth of the camera, as well as the positioning accuracy of the high-precision displacement sensors used for calibration.

In future work, we plan to extend our experiments to simultaneously quantify both longitudinal and transverse displacements using multimode fiber probes. This direction is

expected to play an important role in three-dimensional displacement metrology and in reconstructing three-dimensional microstructures at the nanoscale.

In response to your suggestion, we have added related discussions on transverse displacement metrology in the main text and supplemented the outlook section with feasible technical pathways and implementation schemes, as follows:

(From page 14, line 461)

need for a vacuum environment. In future studies, it is expected that multimode fiber ruler and
hypersurface technology can be used to combine longitudinal displacement measurement with
lateral displacement measurement, which will enable the reconstruction of micro-three-dimensional
surface profiles. On the other hand, by addressing the challenges associated with preparing datasets

(From page 6, line 205)

below the $\lambda/2$ level. This rapid phase variation, or phase singularity, is characteristic of speckle
superoscillation⁴⁵, which has been demonstrated to possess sub-nanometer displacement resolution
capability³³. We believe that the multiple interference effects between the numerous different mode
wavefronts within the multimode fiber lead to such subwavelength free-space singular light fields,
enabling measurements beyond the traditional diffraction limit.[←]

Minor issues:

Comment 9: In the description to Fig. 6c, the SSIM is claimed to be 0.64. Here it should not be a single value but a range. Or simply use 'averaged SSIM' for clarity? If the authors didn't crop and compress the input images, will SSIM be increased?

Response:

Thank you for your valuable suggestions regarding the description in our manuscript. We fully agree with your point that the SSIM value here should not be presented as a single number, but rather as an averaged result. Since SSIM evaluates the structural similarity between each individual image and its reconstruction, different test images naturally yield different SSIM values. In Fig. 6c, the SSIM values of the example images are shown below each corresponding panel. The value of 0.64 reported in the main text represents the average SSIM over the entire test dataset. Following your suggestion, we have revised the description in the main text to explicitly emphasize this, making the statement more rigorous and scientifically accurate, as follows:

(From page 12, line 393)

demonstrate that the system can achieve high-quality imaging. The average SSIM between the
reconstructed and detected patterns achieves 0.64. Additionally, after completing network training,
it can reconstruct patterns distinct from the training set types. The digits, letters, and icons in Fig.
6c demonstrate test results achieved after training on the ImageNet dataset, showcasing the
network's robust generalization capabilities. We believe that nano-displacement detection fiber

For your second question, the SSIM would not increase significantly without cropping and compression. In fact, the cropping operation was introduced to remove irrelevant background and low-signal regions. As shown in the Supplementary Materials, the comparison between the diffuse speckle field and the conventional objective-focused speckle field demonstrates that even after cropping parts of the speckle field, the image reconstruction quality and SSIM can still be well maintained. The same conclusion also holds for displacement measurements.

The compression was applied mainly to reduce computational cost. The current compressed input size of 128×128 does not cause a significant drop in SSIM. Indeed, when we tested a higher input resolution of 256×256 , the reconstructed image quality and SSIM did not show noticeable improvement, as illustrated below.

[editorial note: third party material redacted]

Fig. R43 Image reconstruction results of speckles with 256×256 pixels.

On the other hand, as also demonstrated in the Supplementary Materials, further compression to smaller input sizes does lead to a reduction in SSIM, though the reconstruction retains a certain level of imaging capability.

Comment 10: In Fig. 2f row 1, check if the displacement 0.5 mm or 5 mm. The data in the figure, figure caption and main text are not consistent.

Response:

Thank you for pointing out the inconsistency regarding the displacement value in Fig. 2f (row 1) between the panel, the figure caption, and the main text. In the original manuscript the example displacement was 0.5 mm. In the revised manuscript we have added higher-resolution simulation results (shown below). Following your suggestion, we have carefully verified and corrected the figure, legend, caption, and the corresponding descriptions in the main text to ensure complete consistency. The manuscript and supplementary materials have been updated accordingly.

(From page 4, line 147)

Fig. 2a shows the physical simulation model. Fig. 2b presents the input optical field chosen for
 the simulation. Since the output speckle field is also influenced by the two-dimensional spatial
 structure of the detected target, we selected a wafer micrograph, which contains both high-frequency
 and low-frequency structures, to excite higher-order modes and obtain universal results. Fig. 2c and
 2d show the fiber input facet and field at the core area after free space transmission and fiber-end-
 ball transformation, respectively. We performed mode decomposition on the input optical field and
 selected the first 1000 transverse mode components for reconstruction, as shown in Fig. 2e. It can
 be observed that Fig. 2e contains most of the structures from Fig. 2d. We further analyze the energy
 percentage of the mode components in the input optical field and their relationship with the
 displacement. The first 1000 modes are divided into 10 groups in sequential order, with each group
 containing 100 modes. The intensity percentage of each group is the sum of the intensity percentages of
 the 100 modes in the group. As the detected displacement increases from 0 mm to 30 mm, it can be
 found from Fig. 2f that the share of the fundamental mode component gradually increases, and more

(From page 5, line 165)

Comment 11: The U_1 in equation(S2) should be U_{1F} .

Response:

We sincerely appreciate the reviewer's careful reading and constructive comments. As you pointed out, in Eq. S2 of the Supplementary Materials, the notation was incorrect: the term should be U_{1F} , which denotes the Fourier spectrum, rather than U_1 . The revised content is provided below:

(Supplementary Information, from page 3, line 64)

61	The spectrum of the light field U_2 in front of the fiber-end-ball is obtained by multiplying U_{1F}
62	with the transfer function:↵
63	↵
64	$U_{2F} = U_{1F} \times H_{AS-Air}$ (S2)↵

Comment 12: H_{AS} should use a different quantity name in (S8) to differentiate from its counterpart in (S3).

Response:

We sincerely appreciate the reviewer's suggestion regarding the notation of subscripts. We fully agree with this point. Although both Eq. S3 and Eq. S8 present angular spectrum transfer function, they have different physical meanings and should therefore be distinguished by subscripts. Specifically, in Eq. S3, H_{AS} represents the angular spectrum transfer function describing the light field propagation from the target pattern, through free-space transmission, to the front surface of the fiber ball lens. We have accordingly revised the notation to H_{AS-Air} . In contrast, in Eq. S8, H_{AS} characterizes the angular spectrum transfer function describing the light field propagation from the front surface of the fiber ball lens, through transmission inside the ball lens, to the rear surface (the input facet of the multimode fiber). We have revised this notation to $H_{AS-Ball}$.

These updated notations and subscripts more intuitively convey their respective physical meanings and clearly distinguish between the two cases. We have revised the relevant equations and descriptions in the supplementary material as follows:

(Supplementary Information, from page 3, line 64)

$U_{2F} = U_{1F} \times H_{AS-Air}$ (S2)↵
↵
$H_{AS-Air} = \exp\left(ikz\sqrt{1 - (\lambda f_x)^2 - (\lambda f_y)^2}\right)$ (S3)↵
↵
Where H_{AS-Air} represents the angular spectrum transfer function describing the light field
propagation from the target pattern, through free-space transmission, to the front surface of the
fiber ball lens, k is the wave number, z is the distance between the DMD and the fiber-end-ball,
λ is the wavelength, and f_x and f_y are the spatial frequency components in the x and y directions.↵

(Supplementary Information, from page 4, line 91)

$U_{4F} = U_{3F} \times H_{AS-Ball}$ (S7)↵
$H_{AS-Ball} = \exp\left(ikd\sqrt{1 - (\lambda f_x)^2 - (\lambda f_y)^2}\right)$ (S8)↵
↵
Where $H_{AS-Ball}$ characterizes the angular spectrum transfer function describing the light field
propagation from the front surface of the fiber ball lens, through transmission inside the ball lens, to the
rear surface (the input facet of the multimode fiber).↵

Comment 13: In Fig. S13, the labeled ball lens diameter should be 800 μm according to the information provided.

Response:

We sincerely appreciate the reviewer's careful reading and constructive comments. As correctly pointed out, the diameter of the fabricated multimode fiber ball lens is 800 μm . In the original figure, it was mistakenly labeled as 660 μm , which in fact corresponds to the cladding diameter of the multimode fiber used before fabricating the ball lens (Nufern, 200/220/660 μm). This parameter was inadvertently included in Fig. S13 (as Fig. S22 in revised Supplementary Information). We believe that the corrected labeling will more clearly present the micrograph of the multimode fiber probe inserted into a capillary glass tube. The revised supplementary material is provided as follows:

(Supplementary Information, from page 44, line 1075)